

# A new genus and species of frog from the Kem Kem (Morocco), the second neobatrachian from Cretaceous Africa

Alfred Lemierre[1] and David C. Blackburn[2]

[1] Département Origine et Evolution, UMR 7207, Centre de recherche en Paléontologie, CNR/Sorbonne Université/MNHN, Muséum national d'Histoire naturelle, Paris, France

[2] Department of Natural History, Florida Museum of Natural History, University of Florida, Gainesville, FL, United States of America

## ABSTRACT

Neobatrachia, a clade representing the majority of extant anuran diversity, is thought to have emerged and diversified during the Cretaceous. Most of the early diversification of neobatrachians occurred in southern Gondwana, especially the regions that are today South America and Africa. Whereas five extinct neobatrachians have been described from the Cretaceous of South America in the last decade, only one is known from Africa. This difference in the known extinct diversity is linked to the lack of well-preserved specimens, understudy of fragmentary remains, and lack of known Cretaceous sites in Africa. Study of fragmentary anuran remains from Africa could allow for the identification of previously unknown neobatrachians, allowing for a better understanding of their early diversification. We reanalysed several previously described anuran specimens from the well-known Kem Kem beds, including using CT-scanning. Through our osteological study, we determined that several cranial bones and vertebrae represent a new hyperossified taxon for which we provide a formal description. Comparison to other hyperossified anurans revealed similarities and affinity of this new taxon with the neobatrachians *Beelzebufo* (extinct) and *Ceratophrys* (extant). Phylogenetic analyses supported this affinity, placing the new taxon within Neobatrachia in an unresolved clade of Ceratophryidae. This taxon is the oldest neobatrachian from Africa, and reveals that neobatrachians were already widespread throughout southern Gondwana during the earliest Late Cretaceous.

## INTRODUCTION

The Cretaceous is a key period in anuran evolution and diversification including the emergence of major extant clades such as Neobatrachia and Pipidae (*Frazão, Da Silva & Russo, 2015*; *Feng et al., 2017*). The breakup of the Western Gondwana palaeocontinent during the Late Jurassic and Early Cretaceous (*McLoughlin, 2001*; *Blakey, 2008*)—leading to the creation of the Central and Southern Atlantic Oceans—may have contributed to the early diversification of the Neobatrachia, just as it likely did for the Pipidae (*Frazão, Da Silva & Russo, 2015*; *Feng et al., 2017*). Several neobatrachian taxa have been described in the last decade from the Cretaceous beds of South America (*Báez, Moura & Gómez, 2009*; *Báez et*

Corresponding author
Alfred Lemierre,
alfred.lemierre@edu.mnhn.fr

*al., 2012*; *Báez & Gómez, 2018*; *Agnolin et al., 2020*), contributing to a better understanding of early diversification of the Neobatrachia. Unfortunately, the fossil record of Neobatrachia is scarce for the Cretaceous of Africa and includes only a single described taxon: *Beelzebufo ampinga* (*Evans, Jones & Krause, 2008*) from the Cretaceous of Madagascar (*Evans et al., 2014*). However, the lack of both study and sampling is not limited to either African Cretaceous outgroups or extinct Neobatrachia. In general, there are few well-preserved and identifiable anuran fossils in Africa, with numerous sites yielding only few and fragmentary remains (*e.g.*, *De Broin et al., 1974*; *Báez & Werner, 1996*; *Rage, 2008*; *Gardner & Rage, 2016*) that are not easily incorporated into phylogenetic analyses. This contrasts with South American neobatrachians, several of which are known from well-preserved and mostly articulated specimens preserving much or all of the skeleton (*Báez et al., 2012*). In Africa, only a handful of sites contain enough fragmentary fossils referred to the same taxon to allow for comparisons to other frogs and inclusion in phylogenetic analyses (*Evans, Jones & Krause, 2008*; *Evans et al., 2014*). These few sites are critical to filling the gap in the fossil record of Neobatrachia and central to understanding their early diversification in Africa.

The Kem Kem beds of Morocco (Cretaceous, 100–95 Ma; *Ibrahim et al., 2020*) are known for their rich terrestrial vertebrate fauna with numerous dinosaurs, fishes, sharks, turtles, and crocodiles (*Zouhri, 2017*). This fauna has been studied extensively in recent decades (*Ibrahim et al., 2020*) but there is only a single study of its amphibians. *Rage & Dutheil (2008)* provided evidence for three different anurans, including one pipid that they described as *Oumtkoutia anae* based on a neurocranium, as well as two indeterminate non-pipid anurans based on postcranial remains (*Rage & Dutheil, 2008*). They attributed several cranial fragments to an undescribed species (mainly based on relative size of the cranial and postcranial elements) with an ornamented and hyperossified skull, one of the earliest known from the Cretaceous of Africa. A decade ago, *Agnolin (2012)* described a neobatrachian taxon (Calyptocephalellidae) from the Late Cretaceous of Argentina and reviewed several Gondwanan anurans with hyperossified skulls. In that study, *Agnolin (2012*: 156) included Kem Kem fossils which he referred to Calyptocephalellidae based on cranial and postcranial characters. Because several subsequent studies (*Báez & Gómez, 2018*; *Muzzopappa et al., 2020*) highlighted anatomical and analytical errors in *Agnolin (2012)*, attribution of the Kem Kem fossils to the Calyptocephallelidae is questionable. Because *Agnolin (2012)* considered all of the "indeterminate" anuran remains from the Kem Kem Formation to be a single taxon in his study, several characters supporting the affiliation of these fossils with the Neobatrachia are based on postcranial elements that are not clearly referable to the hyperossified cranial elements. Further, because *Agnolin (2012)* did not include the Kem Kem fossils in his phylogenetic analysis, their relationships were never formally tested. Revaluation of the anatomy and phylogenetic affinities of this hyperossified Kem Kem frog may be important for deciphering the early diversification of neobatrachians during the Lower Late Cretaceous of Gondwana and filling a notable gap in the fossil record of African anurans.

Here, we use microcomputed tomographic scans (MicroCT scans) to provide new information about the anatomy of the hyperossified Kem Kem frog. These new data allow for a more complete anatomical study of this taxon, comparisons to other Cretaceous

anurans, and a phylogenetic analysis to estimate its relationships. We describe this material as a new genus and discuss its importance for understanding neobatrachian diversification in Gondwana during the Cretaceous.

## Geological context

The specimens were collected in 1995 during an expedition organized by the University of Chicago and the Service géologique du Maroc at four different localities near Taouz and Oum Tkout (OT1c, TD1, TZ8a1 and TZ8a2 from *Dutheil, 1999*) from the Kem Kem beds (*Ettachfini & Andreu, 2004*; *Cavin et al., 2010*). The term "Kem Kem beds" (*Sereno et al., 1996*) refers to a large escarpment extending across southeastern Morocco, near the Morocco-Algerian border (*Ibrahim et al., 2020*: figs. 1A and 1C), with numerous exposures along its length. More recently, these beds have been referred to as the Kem Kem group (*Ibrahim et al., 2020*), containing two formations: the Gara Sbaa and the Douira Formations. The anuran specimens discussed here were recovered from layers that can be correlated to the Douira Formation of the Kem Kem group (upper part of the Kem Kem; *Ibrahim et al., 2020*). The Douira Formation (as well as the Gara Sbaa Formation) has been correlated to the Bahariya Formation in Egypt (*Sereno et al., 1996*; *Cavin et al., 2010*), which is dated to the Early Cenomanian (*Cavin et al., 2010*). The Kem Kem group is topped by marine sediments correlated to the Cenomanian-Turonian transition (*Cavin et al., 2010*). Other analyses have confirmed the Cenomanian age (*Ibrahim et al., 2020*) and considered the Kem Kem group a single continuous deposit sequence from 100 to 95 Ma. The boundary between the Gara Sbaa and the Douira Formations is dated to 96 Ma and linked to the Mid-Cenomanian Event (*Ibrahim et al., 2020*). The Douira Formation—and the anuran specimens discussed here—are thus dated to the middle Cenomanian, approximately 96 to 95 Ma (*Ibrahim et al., 2020*).

The Douira Formation contains strata that show a marine influence that increases over time. The deposits in the lower part of the formation, composed of sandstones and mudstones, are consistent with a river delta, whereas the deposits in the upper part, composed of interbedded mudstone with claystone, are characteristic of coastal and sabkha environments (see *Ibrahim et al., 2020* for a complete description). There is no indication of whether the materials came from either lower or upper part of the Douria Formation.

## MATERIAL AND METHODS

The anuran fossils are curated in the vertebrate palaeontology research collection of the University of Chicago. We generated MicroCT scans at the University of Florida's Nanoscale Research Facility using a Phoenix v|tome|x M (GE Measurement & Control Solutions, Boston, MA, USA). Voltage and current were customized for each specimen to balance resolution and intensity contrast; scanning parameters are included in the metadata associated with the scans on MorphoSource. The X-ray images were converted into tomogram slices using GE's reconstruction software datos|x (see Table S1 in Data S1). Each stack of slices produced was imported in the 3D reconstruction software Mimics 21.0 (Materialise, Leuven, Belgium); before importation, slices were cropped to remove empty spaces. To further decrease the data size, the slices were converted from 16 bits to 8 bits.

The resulting slices have an image resolution of 1,580 × 2,144 pixels and a voxel size of 5.7 µm for the volume size. 3D models were produced by segmenting each element using the 'thresholding' function (using the contrast on greyscale images). A 3D model of the endocast was produced by segmenting each element using the "add" function. We used the same voxel resolution of 5.7 µm, with a smoothing factor of 3 for one iteration, to homogenize the model resulting from the segmentation. Data produced by segmentation were exported in the software 3matic 9.0 as separate files (see Table S1 in Data S1).

The electronic version of this article in Portable Document Format (PDF) will represent a published work according to the International Commission on Zoological Nomenclature (ICZN), and hence the new names contained in the electronic version are effectively published under that Code from the electronic edition alone. This published work and the nomenclatural acts it contains have been registered in ZooBank, the online registration system for the ICZN. The ZooBank LSIDs (Life Science Identifiers) can be resolved and the associated information viewed through any standard web browser by appending the LSID to the prefix http://zoobank.org/. The LSID for this publication is: urn:lsid:zoobank.org:pub:DCACD333-53AA-4A6D-A0F0-9F9C180F0DDC. The online version of this work is archived and available from the following digital repositories: PeerJ, PubMed Central SCIE, and CLOCKSS.

## Phylogenetic analyses

Our data matrix includes 88 taxa and 150 morphological characters (62 cranial and 75 postcranial characters, 12 from the hyobranchial apparatus, and one from soft-tissues) and is derived from that of *Lemierre et al.* (*2021*; see Appendices S1, S2, S3). We added two extinct hyperossified neobatrachian taxa (the new taxon described below from the Kem Kem, and *Hungarobatrachus szukacsi*) to test their affinities. *Hungarobatrachus szukacsi* (*Szentesi & Venczel, 2010*) has recently been included in a reduced phylogenetic analysis (*Venczel, Szentesi & Gardner, 2021*) and is considered a neobatrachian. It is the oldest neobatrachian outside of Gondwana and essential to understand the diversification of the clade during the Cretaceous. These new taxa were scored from observation on 3D mesh files created for this study based on segmenting newly generated MicroCT scans (see above) and from literature (*Szentesi & Venczel, 2010*; *Venczel, Szentesi & Gardner, 2021*).

All analyses were performed using TNT v.1.5 (*Goloboff & Catalano, 2016*). All analyses were conducted with cline (also called multi-state) characters ordered (characters 3, 9, 10, 14, 26, 34, 51, 52, 68, 93, 112, 121, 124, 125 and 126). Cline characters were ordered as several studies (*Rineau et al., 2015*; *Rineau, Bagils & Laurin, 2018*) showed that analyses using ordered morphocline characters outperformed analyses using unordered characters, even when the ordering scheme is wrong (*Rineau, Bagils & Laurin, 2018*). Analyses consisted of heuristic searches with 1,000 random addition sequences of taxa, followed by tree bisection reconnection (TBR) branch swapping, withholding 10 trees per repetition. The final trees were rooted using *Ascaphus* Stejneger 1899 (Ascaphidae, Anura) and a strict consensus was created. Node supports were evaluated using Bremer support and standard nonparametric bootstrapping, with searches of 1,000 replicates and collapsing groups below 5% frequency.

Because the phylogeny resulting from the above analysis is strongly at odds with relationships inferred from those inferred with molecular genetic data, we performed an additional analysis using a constraint tree reflecting a consensus of recent molecular phylogenetic analyses. This included constraining the backbone of the tree to reflect early divergences in anuran evolution, as well as large-scale patterns of relationships within the two major clades of Neobatrachia (Hyloidea, Ranoidea). Within Hyloidea, we constrained four clades: Calyptocephalellidae, Neoaustrarana (*Feng et al., 2017*; *Streicher et al., 2018*), the genus *Telmatobius* Wiegmann, 1834 as monophyletic, and a clade representing all other hyloids. Within Ranoidea, we constrained three clades: Afrobatrachia, Microhylidae, and Natatanura. We did not constrain the placement of any extinct taxa and we also left relationships within constraint clades (*e.g.*, Pelobatoidea, Myobatrachoidea, Natatanura) as polytomies so that relationships within them could be inferred by our morphological data. This constraint tree (available in the Supplemental Materials) was generated by hand and represents a broad-scale consensus of phylogenetic relationships presented in recent phylogenomic analyses for most frog families (*Feng et al., 2017*: fig. 1; *Hime et al., 2021*: fig. 1) and those specific to hyloids (*Streicher et al., 2018*: fig. 6) and ranoids (*Yuan et al., 2018*: fig. 2).

The anatomical terminology used herein is based on *Roček (1981)*, *Sanchiz (1998)*, and *Biton et al. (2016)* for cranial features, *Sanchiz (1998)* for postcranial ones, *Gómez & Turazzini (2021)* for humerus anatomy, and *Gómez & Turazzini (2016)* for ilium anatomy. Anatomical terminology for cranial nerves follows *Gaupp (1896)*.

## RESULTS

### Systematic Paleontology

> *Anura* Duméril, 1804
> *Neobatrachia* Reig, 1958
> *Cretadhefdaa* gen. nov.

### Type (and only known) species
*Cretadhefdaa taouzensis sp. nov.*
  *CRETADHEFDAA TAOUZENSIS* sp. nov.

### Holotype
UCRC-PV94, posterior portion of the skull preserving co-ossified and incomplete frontoparietals, parasphenoid, and the prooticooccipital (the co-ossified prootics and exoccipitals sensu *Roček, 1981*).

### Type locality
TD1, near the city of Taouz in southeastern Morocco (see *Dutheil, 1999* for more information on Kem Kem localities).

### Stratigraphic range
Middle Cenomanian (96–95 Ma).

### Referred materials

One incomplete squamosal from TD1 (UCRC-PV95); one incomplete maxilla from Tz8a1 (UCRC-PV96); three incomplete presacral vertebrae, two from TD1 (UCRC-PV97–98) and one from Tz8a1 (UCRC-PV101); one incomplete sacral vertebra from OT1c (UCRC-PV103).

### Etymology

The genus nomen *Cretadhefdaa* is a combination of the word Cretaceous and a transliteration of the Arabic word *dhefdaa* (also sometimes written as *dheftha* or *thedfaa*), meaning "frog". The specific epithet *taouzensis* recognizes the type locality, Taouz.

### Diagnosis

A neobatrachian anuran with a hyperossified skull differing from all other anurans by the following unique combination of characters: (1) frontoparietals coossified, lacking a midline suture, and covered in ornamentation of pits and ridges; (2) frontoparietals bearing a smooth occipital flange; (3) no incrassatio frontoparietalis on the ventral surface of the frontoparietals; (4) presence of a deep, groove-like central recess on the posterodorsal surface of the braincase to each side of the foramen magnum, and housing the foramen for the arteria occipitalis.

    *Cretadhefdaa* can be differentiated from *Beelzebufo* in (1) having a smooth occipital flange on the posterior margin of the frontoparietals; (2) having a ventral extension of maxillary ornamentation on the pars dentalis and (3) lacking an ornamented table sitting atop neural spine of anterior presacral vertebrae. *Cretadhefdaa* can be differentiated from *Baurubatrachus* in (1) having a fully ossified dorsal margin of the foramen magnum; (2) lacking a distinct palatine shelf of the maxilla; (3) having a smooth occipital flange on the posterior margin of the frontoparietals; and (4) having a slender and shorter neural spine on presacral vertebrae. *Cretadhefdaa* can be differentiated from *Calyptocephalella satan* *Agnolin, 2012* in (1) lacking a distinct shelf on the maxilla; (2) having a smooth occipital flange on the posterior margin of the frontoparietals; (3) lacking median suture between frontoparietals; and (4) having weakly expanded sacral transverse processes. *Cretadhefdaa* can be differentiated from *Hungarobatrachus* in (1) lacking an incrassatio frontoparietalis on the ventral surface of the frontoparietals; (2) having the arteria occipitalis foramen within a deep recess; and (3) lacking a distinct palatine shelf of the maxilla. Diagnosis for the species is same as for the genus.

### Description of the holotype (UCRC-PV94)
#### *Osteological description*

UCRC-PV94 is the preserved posterior region of the skull of *Cretadhefdaa*. All bones are co-ossified and the sutures between prooticooccipitals and the frontoparietals are difficult to discern (Figs. 1A–1G).

    The posterior portion of the frontoparietals is preserved. The two frontoparietals are coossified to one another, and no suture is visible on the frontoparietal table (Fig. 1A). The frontoparietal table is large and covered in an ornamentation of pits and ridges. The posterior margin of the frontoparietals is flanked by a large occipital flange that

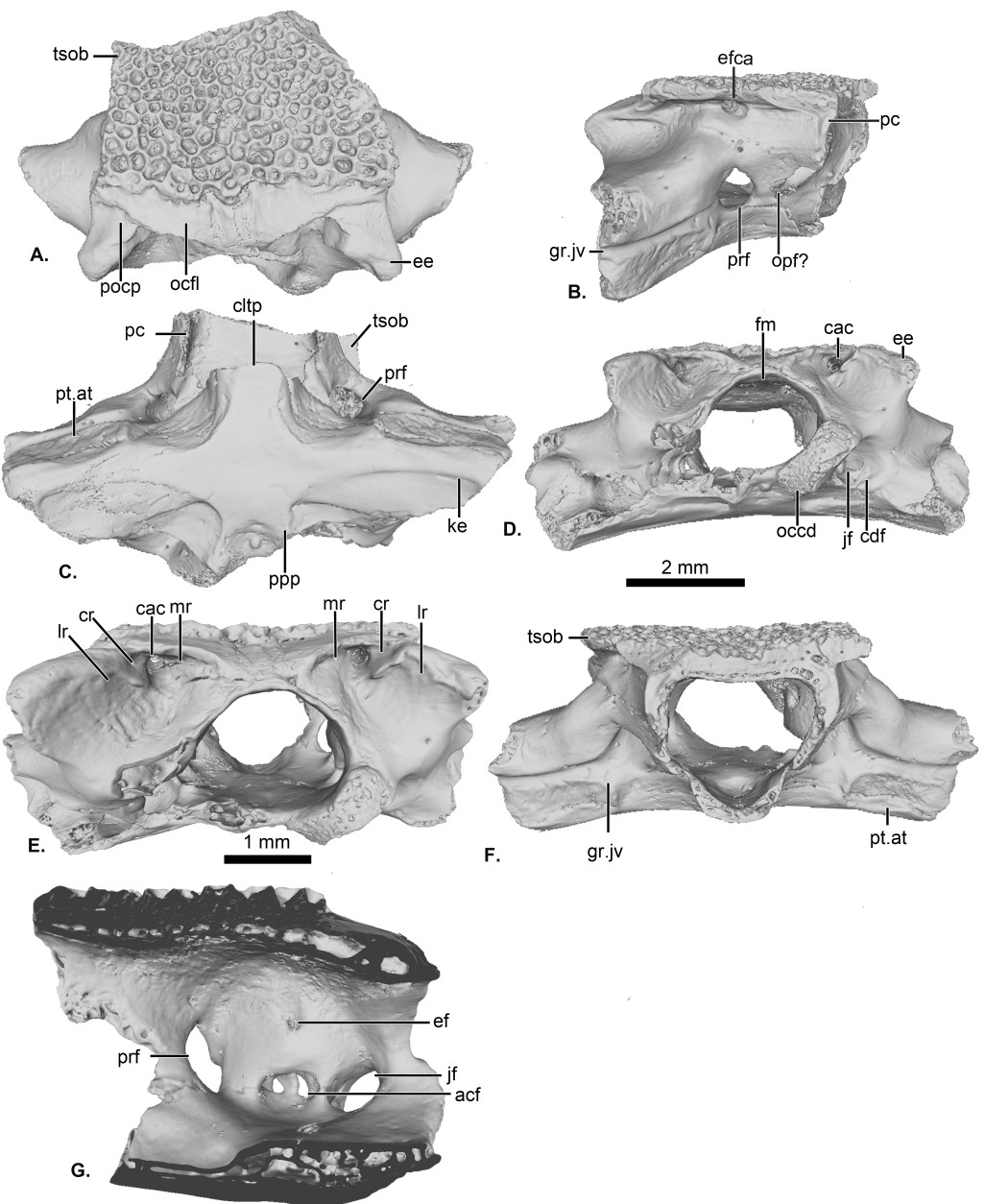

**Figure 1** **UCRC-PV64, holotype of *Cretadhefdaa*.** Incomplete braincase in (A) dorsal; (B) right lateral; (C) ventral; (D) posterior; (E) posterodorsal view with a close up on the recesses system; (F) anterior and (G) medial views. Abbreviations: acf?, fused acoustic foramina; cac, canal for arteria occipitalis; cdf, condyloid fossa; cltp, cultriform process; cr?, central recess; ee, epiotic eminence; ef, endolymphatic foramen; efca, exit foramen for arteria orbitonasalis; fm, foramen magnum; gr.jv, groove for the jugular vein; jf, foramen jugulare; ke, median keel; lr, lateral recess; mr, medial recess; occd, occipital condyle; opt?, optic foramen; pc, pars contacta; pocp, paraoccipital process; ppp, posterior process of the parasphenoid; prf, prootic foramen; pt.at, pterygoid attachment area; tsob, tectum supraorbitale.

lacks ornamentation (Fig. 1A). The paraoccipital processes are reduced and fused to the underlying epiotic eminence (prominentia circularis ducti posterior of *Roček & Lamaud, 1995*),  and the posterior process of the frontoparietals is not distinct (Fig. 1A). There is no pineal foramen visible. In lateral view, the preserved portion of the pars contacta is a straight vertical lamina (Fig. 1B). In ventral view, the frontoparietal table extends lateral to the pars contacta into a tectum supraorbitale, but its full extent is unknown because it is broken (Figs. 1A and 1B). There is no visible frontoparietal incrassation on the ventral surface of the frontoparietals (*i.e.*, there is no imprint of the dorsal surface of the endocranium). The absence of frontoparietal incrassation could linked to the coossification of the tecta and thickening of this region of the frontoparietals (Z. Roček, pers. com.). In posterior view, the boundary between the frontoparietals and prooticoccipitals bears a series of deep recesses (Figs. 1D and 1E). The recesses are located between the tall epiotic eminence and the posterior margin of the frontoparietal table and appear to form a single large, deep groove on each side of the braincase. However, three different recesses can be distinguished within each groove (medial, central, and lateral recesses in Fig. 1E) that are each separated by well-defined ridges. Both the lateral and medial recesses are shallow, whereas the central recess is deep and houses a large, circular foramen for the arteria occipitalis. The foramen for the arteria orbitonasalis is visible on each lateral surface of the frontoparietal, ventral to the lateral extension of the table (Fig. 1B).

The posterior region of the parasphenoid is preserved. The cultriform process is broken, preserving only its base. The alae are large and cover the ventral surface of the otic capsules. In ventral view, the alae bear a median keel on its surface, extending from its lateral margin to and slightly curving towards the posterior process of the bone (Fig. 1C). The posterior process is divided into two well-separated small extensions, oriented posterolaterally. These expansions are fused to the base of the occipital condyles (Fig. 1C).

The prootic and exoccipital are coossified into a single prooticooccipital complex without a visible suture. Each prooticooccipital is co-ossified to the other along their medial margins, as well as to the frontoparietals (dorsally) and parasphenoid (ventrally). In dorsal view, the epiotic eminence is large, forming a broad lamina (Figs. 1A and 1D). The dorsal surface of the prootic is smooth. The crista parotica is not fully persevered, but likely had an ossified lateral expansion. There is no trace of an articulation facet for the squamosal on the preserved portion of the prooticoccipital (Fig. 1A). In anterolateral view, a large prootic foramen is present on the anterior surface of the prooticoccipital (Figs. 1B and 1G), and is fully enclosed in bone. In lateral view, anterior to the prootic foramen, a notch is visible on the anteriormost bony margin of the braincase (Fig. 1B) and might represent the posterior portion of the optic foramen. In anterior view, a well-delimited, narrow groove, likely for the jugular vein, extends from a large depression at the border of the prootic foramen to the lateral margin of the prootic (Fig. 1F). Beneath this groove, a large depression is present from the lateral margin of the prootic to the midpoint of its anterior surface. This is likely an articular facet for the medial ramus of the pterygoid. In posterior view, the left occipital condyle is missing (Fig. 1D), but the right occipital condyle is slightly ventrolateral to the large foramen magnum (Fig. 1D). The occipital condyle obscures the foramen jugulare that remains partially visible laterally (Fig. 1D). In

medial view, several foramina are visible in the wall of the braincase. The posteriormost opening is the foramen jugulare (Fig. 1G). Separated from the latter foramen by a thin bony pillar, a large opening is present on the lateral braincase wall (Fig. 1G). This opening likely represents the fused acoustics foramina, a fusion that is common in many anurans (Z. Roček, pers. comm.). A similar preservation is also present in the exceptionally preserved *Thaumastosaurus servatus Filhol, 1877* (*Lemierre et al., 2021*).

### Inner ear

The preservation of the endocast of the otic capsule allowed us to segment the otic chamber and semi-circular canals (vestibular apparatus) of *Cretadhefdaa*. The anterior, posterior, and lateral canals are all preserved and clearly identifiable (Figs. 2A and 2B). In anterior view, the base of the anterior canal bears a bulge, containing the anterior ampulla (Fig. 2A). In dorsal view, at the base of both anterior and lateral canals, the bulges contain the anterior and lateral ampullae (Fig. 2C). At the base of the posterior sinus (connecting the lateral and posterior canals), a similar bulge contains the posterior ampulla (Figs. 2B and 2D). In anterior and posterior views, the common crus (superior sinus), connecting the anterior and posterior canals, is well preserved (Figs. 2A–2C). The base of the superior sinus is thick, and is part of the utricle. The utricle forms the ventral portion of the vestibular apparatus. The vestibular apparatus occupies approximately half of total height of the endocast. The auditory region is large and bulbous (Fig. 2), and the posterior region (perilymphatic cistern + sacculus + lagena) occupies most of the endocast.

Within this posterior region, the perilymphatic cistern occupies the posteromedial region (Fig. 2B), while the sacculus occupies the anteriormost portion of this region (Fig. 2A). Lateral to the posterior region, a small region is delimited from the rest of the ventral volume by a slight constriction (Figs. 2A and 2B). This region can be identified as the transverse section through the perilymphatic space close to the fenestra ovalis. Near the perilymphatic cistern, a short and large canal, representing the perilymphatic ducts, opens posteriorly (Figs. 2B and 2D) into the braincase and the condyloid fossa (fused perilymphatic foramina). Another large duct is visible in the medial region of the otic chamber, entering the braincase through the fused acoustic foramina. However, this canal comprises two smaller ducts that are fused medially (Fig. 2) and housed the pathway of the cranial nerve VIII (*Gaupp, 1896*), representing the auditory nerve (*Duellman & Trueb, 1994*). A second medial, smaller duct is visible in the medial region of the vestibular apparatus, leading to the dorsalmost foramen of the lateral wall of the braincase (Fig. 1G). This duct is identified as the endolymphatic duct, leading to the endolymphatic sac that was present in the braincase (*Frishkopf & Goldstein, 1963*; *Duellman & Trueb, 1994*).

### Referred cranial material
#### UCRC-PV95

The specimen is a fragment of a right squamosal preserving part of the lamella alaris (otic plate of *Evans et al., 2014*) and the base of the processus posterolateralis (Fig. 3). The dorsal and lateral surfaces of the bone are covered with an ornamentation made of deep longitudinal pits and ridges in the orbital and lateral region, and deep, nearly circular pits and ridges in the posterior and otic region (Fig. 3A). This ornamentation is slightly

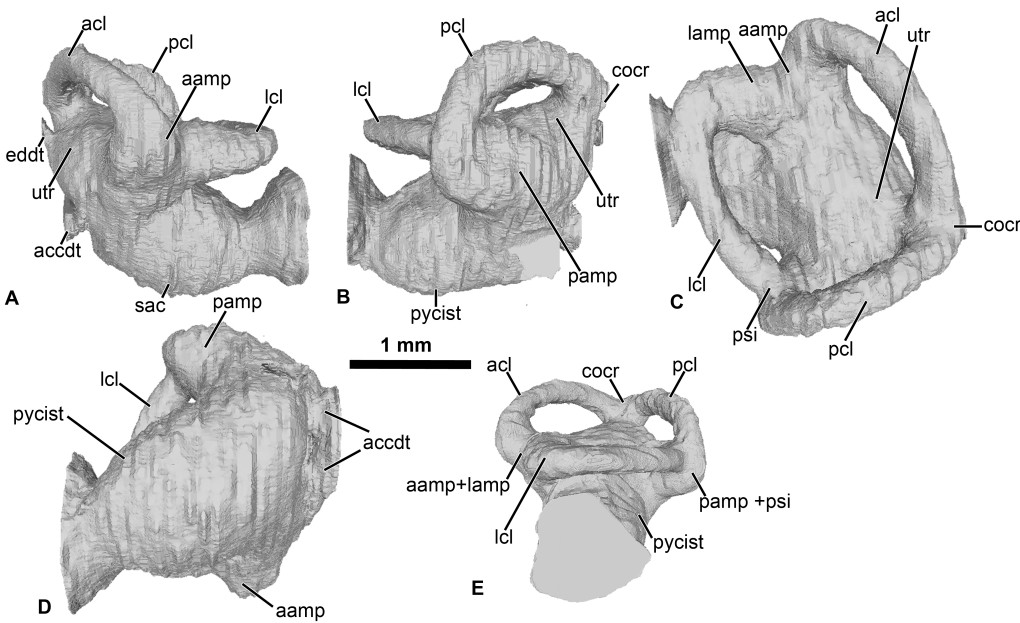

**Figure 2 Internal morphology of the otic capsule of *Cretadhefdaa.*** Endocast of the left otic capsule in (A) anterior; (B) posterior; (C) dorsal; (D) ventral and (E) lateral views. Abbreviations: aamp, anterior ampulla; accdt, transverse section through perilymphatic space containing stat-acoustic nerve; acl, anterior semicircular canal; cocr, common crus; eddt, endolymphatic duct; lamp, lateral ampulla; lch, lateral chamber; lcl, lateral semicircular canal; otch, otic chamber; pamp, posterior ampulla; pcl, posterior semicircular canal; psi, posterior sinus; pycist, perilymphatic cistern; sac, sacculus; utr, utricle.

different from that observed in UCRC-PV94, though it is not uncommon for anuran cranial bones to display variation in ornamentation within an individual (*de Buffrénil et al., 2015*; *de Buffrénil et al., 2016*). Thus, we interpret UCRC-PV95 as belonging to the same taxon as UCRC-PV94. The size of the squamosal is consistent with the size of the braincase (UCRC-PV94), but there is no indication that the two bones belong to the same individual. The lamella alaris is well developed (∼3 mm length preserved, anterior to posterior) with an anterior extension ventrolaterally oriented (Fig. 3B) . Posteriorly, the lamella alaris bears a vertical lamina, likely the base of the processus posterolateralis (Fig. 3C). Near this lamina, on the ventral surface of the lamella alaris, a small broken ridge is present. It might be the base of the ramus paroticus.

### UCRC-PV96

This represents a partial anterior portion of a right maxilla. The maxilla is toothed and its lateral surface is covered in a pits and ridges ornamentation. The ornamentation covers almost all of the lateral surface, save for a thin strip of bone ventrally and its dorsalmost portion. Dorsally, the base of the large processus frontoparietalis is preserved (Figs. 3E and 3F). In medial view, the pars dentalis is straight, with a small sulcus dentalis (Figs. 3F and 3G; also visible in ventral view). The lamina horizontalis is faint, almost non-distinct from the medial surface of the maxilla (Fig. 3F). It forms a small ridge, with a shallow dorsal groove for the palatoquadrate (Fig. 3G). A deep maxillary recess is present medially (Fig.

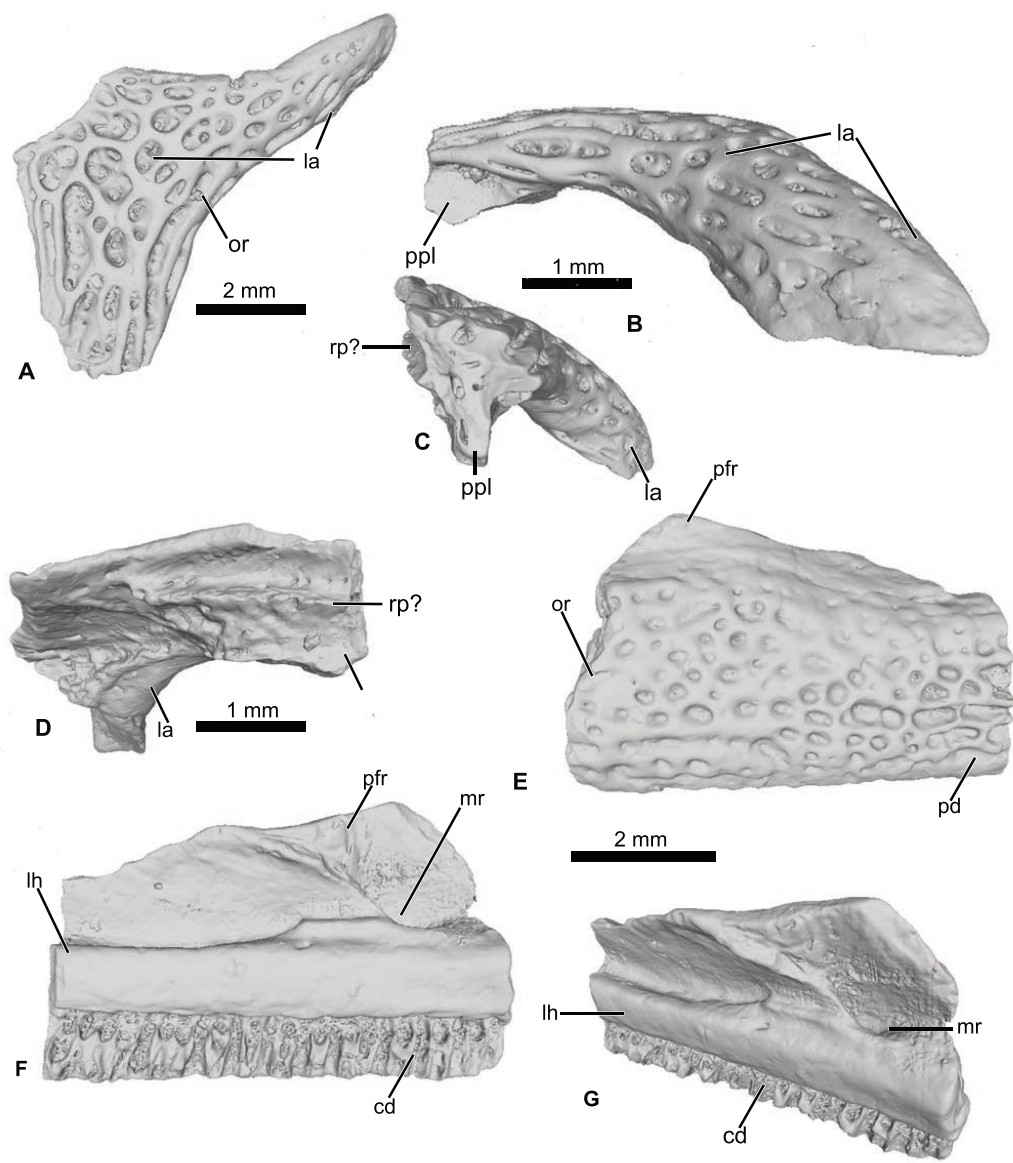

**Figure 3 Cranial elements of *Cretadhefdaa*.** (A–D) UCRC-PV95, incomplete squamosal in (A), dorsal; (B) lateral; (C) posterior and (D) ventral views; (E–G) UCRC-PV94, incomplete maxilla in (E) lateral, (F) medial and (G) dorsomedial views. Abbreviations: cd, crista dentalis; la, lamella alaris; lh, lamina horizontalis; mr, maxillary recess; or, ornamentation; pfr, processus frontalis; ppl, processus posterolateralis; rp?, ramus paroticus ?.

3F). A groove for maxillary nerves extends dorsally from the maxillary recess to the dorsal part of the maxilla. Because only the bases of several teeth are preserved, nothing can be said of the tooth morphology of *Cretadhefdaa*.

## Referred Vertebrae

The four vertebrae attributed to *Cretadhefdaa* all have an anterior cotyle and a posterior condyle, indicating a procoelous condition of the vertebral column. Although the length

of the centrum varies among these specimens (UCRC-PV97, UCRC-PV101 and UCRC-PV103 are shorter than UCRC-PV98), their similar size and the shape of articular facets and zygapophyses suggests that they all represent the same taxon. In addition, the two best preserved vertebrae, UCRC-PV101 and UCRC-PV98, have a similarly shaped low and short neural spine that is oriented posteriorly. In other anurans, there is documented variation in the length of the centra of presacral vertebrae throughout the vertebral column (*Trueb, 1973*; *Duellman & Trueb, 1994*; *Púgener, 2002*; *Evans et al., 2014*; *Lemierre et al., 2021*: fig. 9). We attribute the above cranial elements and these vertebrae to *Cretadhefdaa* because they all represent non-pipid individuals of similar body size (following *Rage & Dutheil, 2008*).

### UCRC-PV97

This specimen is a centrum of a procoelous vertebra, with the neural walls not preserved (Figs. 4A–4C). The centrum is longer than wide (Figs. 4A and 4B). The posterior condyle is large and wide.

### UCRC-PV98

This presacral vertebra is better preserved than UCRC-PV97, with most of the transverse process, one postzygapophysis, and the distal end of the neural spine missing (Figs. 4D–4I). The width of the posterior condyle is the same as that of the vertebral canal. The neural walls are thick, with the base of the transverse processes protruding laterally. In dorsal view, the remnants of the transverse processes are subcylindrical and oriented posteriorly. Each prezygapophysis bears a large flat and ovoid-shaped articular facet that is oriented dorsomedially (Fig. 4F). The medial margin of this articular facet is a sharp, straight lamina constituting the medial end of the dorsal wall of the vertebral canal. The neural spine is low and was likely short, though it is broken distally. The postzygapophysis is long, with an ovoid and flattened articular surface that is oriented ventrally (Fig. 4F). A small posterior lamina connects the neural spine and the medial margin of the postzygapophysis. The centrum is more elongate than UCRC-PV97 (Fig. 4G). In ventral view, the centrum is compressed lateromedially at midlength, giving the ventral surface an hourglass shape (Fig. 4G). In lateral view, a shallow fossa is visible at the midpoint of the vertebra and might be a remnant of a spinal foramen (Figs. 4H and 4I). The elongate centrum indicates that this vertebra is from the mid-column of *Cretadhefdaa*, possibly representing presacral vertebra IV.

### UCRC-PV101

This element is an incomplete presacral vertebra preserving the centrum and neural arch (Figs. 4J–4N). The centrum is short, almost as wide as long. The vertebra is procoelous, with an anterior cotyle and a posterior condyle (Figs. 4J and 4K). The condyle is poorly preserved but seems elongated lateromedially. The prezygapophyses bear a flat articular facet that is oriented dorsomedially (Fig. 4L). In dorsal view, the anterior margin of the neural arch is concave, and a sharp ridge is visible on the dorsal surface of the neural arch, marking the beginning of the neural spine. The neural spine is very short (shorter than the one recovered in UCRC-PV98) and oriented posteriorly. Each postzygapophysis

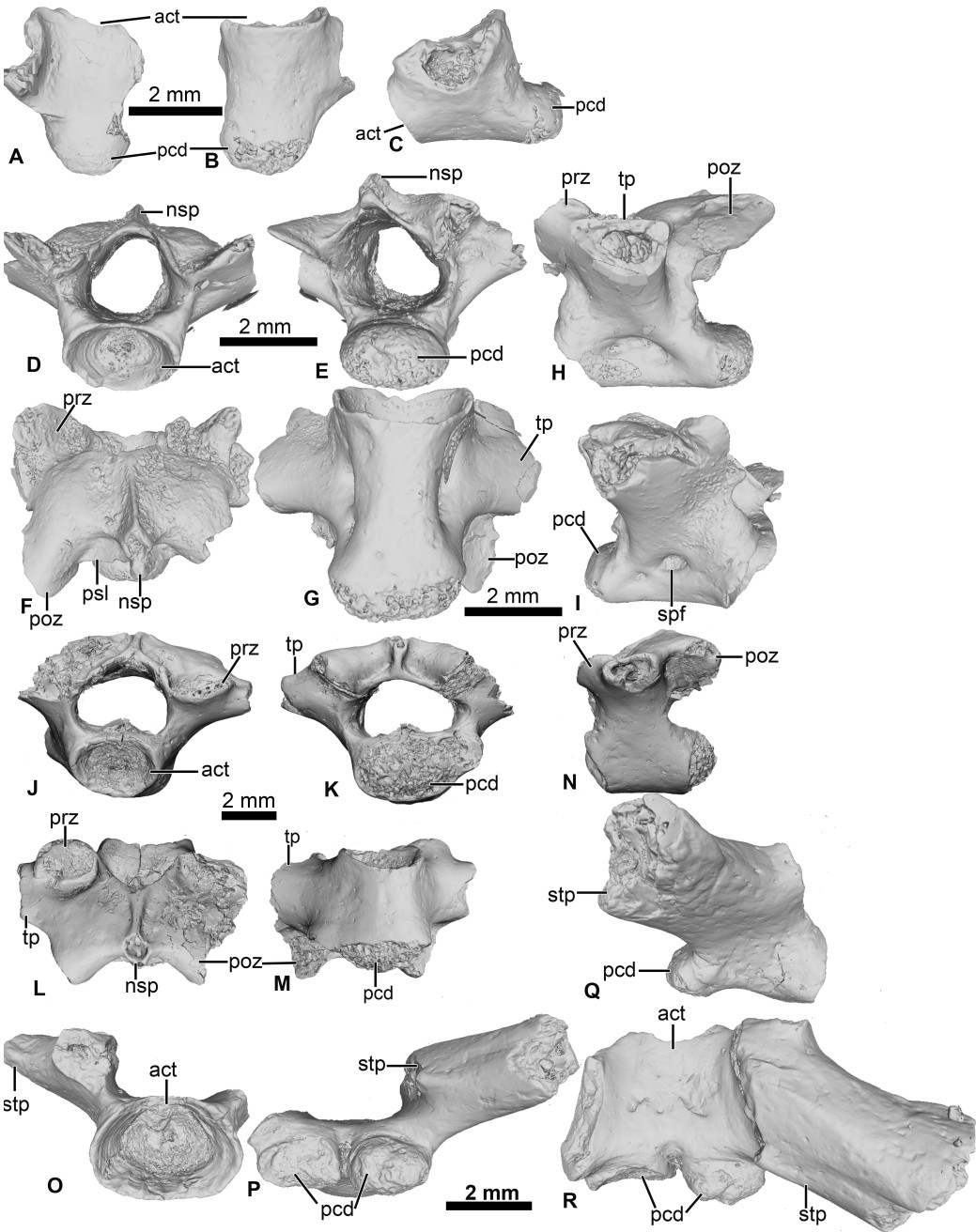

**Figure 4  Vertebral element of *Cretadhefdaa*.** (A–C) UCRC-PV97, presacral centrum in (A) dorsal; (B) ventral and (C) right lateral views; (D–I) UCRC-PV98 incomplete possible presacral vertebra IV in (D) anterior; (E) posterior; (F) dorsal; (G) ventral; (H) left lateral and I right lateral views; (J–N) UCRC-PV101, incomplete possible presacral VIII in (J) anterior; (K) posterior; (L) dorsal; (M) ventral and N left lateral views; (O–R) UCRC-PV103, incomplete sacral vertebra in (O) anterior; (P) posterior; (Q) right lateral and (R) ventral views. Abbreviations: act, anterior cotyle; nsp, neural spine; pcd, posterior condyle; poz, postzygapophysis; prz, prezygapophysis; psl, posterior lamina; spf, spinal foramen; stp, sacral transverse process; tp, transverse process.

bears a flat articular surface that is oriented ventrolaterally. The transverse processes are broken at their bases. The base of these processes is cylindrical in shape and elongate anteroposteriorly, oriented perpendicular to the anteroposterior axis of the centrum (Figs. 4L and 4N). The anteroposteriorly short centrum and the low and posteriorly oriented neural spine indicate that UCRC-PV101 is one of the posterior presacral vertebrae (VI to VIII). The posterior condyle of UCRC-PV101 is similar in size to the anterior cotyle of the identified sacral vertebra (UCRC-PV103) and the inferred position of the prezygapophyses of UCRC-PV103 seems to match the position of the postzygapophyses of UCRC-PV101. UCRC-PV101 might represent the last presacral vertebra (VIII).

### UCRC-PV103

This incomplete sacral vertebra bears an anterior cotyle and two posterior condyles (Figs. 4O–4R). The centrum of UCRC-PV103 is shorter than the other three vertebrae, but the anterior cotyle is similar to those of UCRC-PV97–98 and 101. The two posterior condyles are well separated and are wider than tall, and thus elliptical. The preserved transverse process is posterolaterally oriented and the preserved portion does not expand distally. In lateral view, the sacral transverse process is extended anteroposteriorly, and is not cylindrical or rod-like. The dorsal expansion of the transverse process is visible in dorsal view (Fig. 4R).

### Osteological comparison to hyperossified anurans

Hyperossified (sensu *Trueb, 1973*) ornamented cranial bones occur in both extinct and extant anurans, from pipoids (*Báez & Rage, 1998*; *Trueb, Púgener & Maglia, 2000*) to diverse lineages of neobatrachians, and is a condition that has evolved more than 20 times independently across extant frogs (*Paluh, Stanley & Blackburn, 2020*). Hyperossified cranial elements are known in numerous Cretaceous anurans from both Laurasian and Gondwanan sites (*Jacobs, Winkler & Gomani, 1990*; *Rage & Roček, 2003*; *Roček, 2013*; *Gardner & Rage, 2016*). In the Gondwanan fossil record, Cretaceous hyperossified anurans are known that belong to both the Pipimorpha and Neobatrachia (*Gardner & Rage, 2016*; *Báez & Gómez, 2018*).

### Comparison to non-neobatrachian taxa

Ornamented and co-ossified cranial bones are relatively uncommon in the first four diverging lineages of extant frogs: Leiopelmatoidea, Alytoidea, Pipoidea, and Pelobatoidea. Neither of the two extant leiopelmatoids, *Ascaphus* and *Leiopelma* Fitzinger 1861, exhibit any characteristics unique to hyperossified anuran skulls. Among the extant alytoids, ornamented dermal bones are found only in the genus *Latonia* Meyer 1843 which is known from the Paleogene and Neogene of Laurasia and Africa (*Roček, 1994*; *Roček, 2013*; *Biton et al., 2016*). However, *Cretadhefdaa* differs from *Latonia* in having a foramen for the occipital artery (lacking in *Latonia*) and frontoparietals that fuse with the prooticooccipitals (see *Roček, 1994*: fig. 7). The extinct Gobiatidae from the Cretaceous of Asia (*Roček, 2008*; *Roček, 2013*) also exhibits ornamented dermal bones. However, *Cretadhefdaa* can be differentiated from all Gobiatidae in having fused frontoparietals without a visible suture (frontoparietals not fused or in contact with each other in Gobiatidae), complete fusion of the prootic and

exoccipital (suture visible between the two bones in Gobiatidae; *Roček, 2008*), and presacral vertebrae that are procoelous (amphicoelous in Gobiatidae).

*Cretadhefdaa* can be differentiated from all pipoid anurans in having alae of the parasphenoid that cover the ventral surface of the otic capsules (Fig. 1C). Some members of the Pelobatoidea also have ornamented skull bones, but as an integral part of the bone and not as a secondary exostosis (*Rage & Roček, 2007*; *Roček, 2013*; *Roček et al., 2014*). *Cretadhefdaa* can be differentiated from *Eopelobates* Parker, 1929 in (1) having ornamentation as a secondary exostosis (ornamentation is an integral part of the bones in *Eopelobates*); and (2) lacking anteroposterior expansion of the distal part of the sacral apophyses (*Roček et al., 2014*).

In addition, several fragmentary remains of ornamented maxillae and procoelous vertebrae were recovered in the Cretaceous outcrops of Texas and might represent one the early diverging frog lineages, but the phylogenetic affinities of these fossils remain unclear (*Roček, 2013*).

Two hyperossified taxa of uncertain affinities are known from the Late Cretaceous of North America: *Scotiophryne* Estes, 1969 and *Theatonius* Fox, 1976. *Cretadhefdaa* can be differentiated from *Scotiophryne* in (1) having ornamentation made of pits and ridges (fine beadlike tubercles in *Scotiophryne*; *Gardner, 2008*); (2) having fused frontoparietals without a median suture (frontoparietals not fused in *Scotiophryne*); (3) having a well-delimited lamina horizontalis on the maxilla; and (4) having a well-developed ramus paroticus of the squamosal that articulates with the frontoparietal. *Cretadhefdaa* can be differentiated from *Theatonius* in (1) having teeth on the maxillae (maxillae are edentate in *Theatonius*; *Gardner, 2008*); and (2) having fused frontoparietals without a median suture (frontoparietal fused with a median suture in *Theatonius*; *Gardner, 2008*).

Based on the above comparisons, we exclude *Cretadhefdaa* from the Leiopelmatoidea, Alytoidea, Pipoidea, and Pelobatoidea. The vast majority of extant frog species belong to the Neobatrachia. *Cretadhefdaa* shares with Neobatrachia the presence of well-separated occipital condyles and a bicondylar articulation between the sacrum and urostyle. However, others synapomorphies used to diagnose Neobatrachia (in combination with the two mentioned above), such as the presence of palatines (also called neopalatines in neobatrachians; *Báez, Moura & Gómez, 2009*) cannot be assessed based on the preserved elements of *Cretadhefdaa*.

## Comparison to Cretaceous hyperossified taxa

The best known non-pipimorph ornamented taxon described from the Mesozoic fossil record of Africa is *Beelzebufo ampinga*, from the Maastrichtian of Madagascar (*Evans, Jones & Krause, 2008*; *Evans et al., 2014*). *Beelzebufo* is known from numerous cranial and some postcranial elements. The ornamentation of *Cretadhefdaa*, comprised of pits and ridges, is similar to that of *Beelzebufo*. Both taxa also have a series of three recesses on the posterodorsal surface of the skull, with the foramen for the arteria occipitalis located within the central recess, which is the deepest recess in both taxa (Fig. 5). *Cretadhefdaa* also differs from *Beelzebufo* in having a smooth occipital flange on the posterior region of the frontoparietals. The poor preservation of the tectum supraorbitale of the frontoparietals

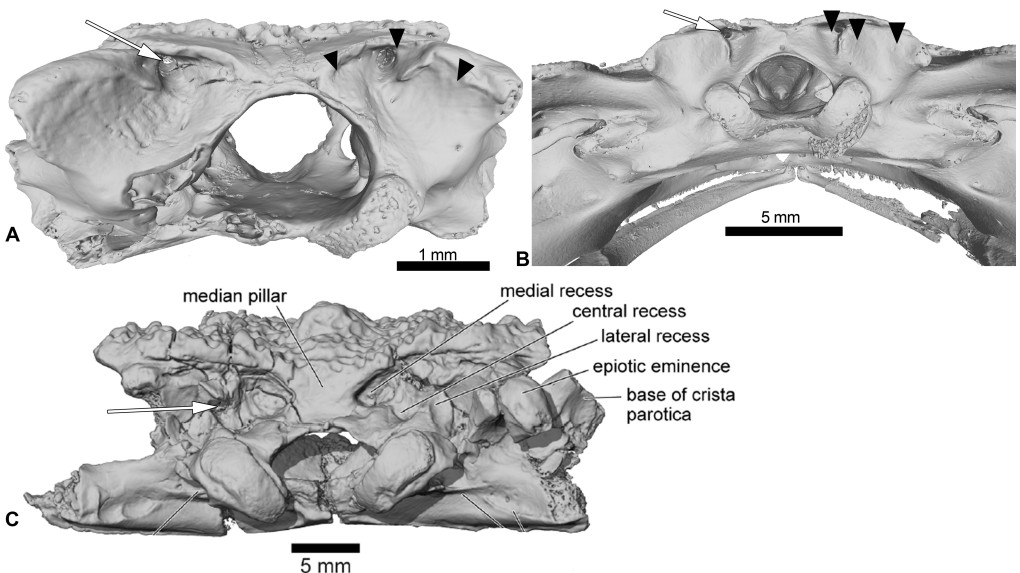

**Figure 5** Comparison between the braincases of *Cretadhefdaa*, *Beelzebufo* and Ceratophryidae. (A) *Cretadhefdaa* in posterior view (UCRC-PV64); (B) *Beelzebufo* braincase in posterior view (taken from (*Evans et al., 2014*): fig. 22C) and (C) braincase of*Ceratophrys aurita* in posterior view (CAS:Herp:84998; MorphoSource ARK: ark:/87602/m4/M16099). Black arrows point to the recesses discussed in the text and white arrows point to the foramen for the arteria occipitalis.

of *Cretadhefdaa* (UCRC-PV94) means that we cannot evaluate whether it is similar to the expansion in *Beelzebufo*, in which the tectum supraorbitale is elongate laterally along on its entire length, covering the lateral region of the braincase (*Evans et al., 2014*). The parasphenoid of *Cretadhefdaa* is similar to that of *Beelzebufo* in having narrow alae (alary process of *Evans et al., 2014*) with a median keel. *Cretadhefdaa* is similar to *Beelzebufo* in lacking a distinct palatine shelf on the medial surface of the maxilla, but differs in having ornamentation of the pars facialis on the lateral surface of the maxilla that extends ventrally to the pars dentalis (the ornamentation ends before the pars dentalis in *Beelzebufo*).

The presacral vertebrae of *Cretadhefdaa* differ from most of those referred to *Beelzebufo* by lacking a well-developed neural spine, and lacking an expanded and ornamented "table" sitting atop the spine (*Evans et al., 2014*: figs. 34–36). In addition, even the shortest neural spine of the posteriormost presacral of *Beelzebufo* is taller than that of any vertebrae that we refer to *Cretadhefdaa* (Figs. 4F and 1L). The sacral vertebra of *Cretadhefdaa* is similar to that of *Beelzebufo* in having two elliptical posterior condyles for the sacro-urostylar articulation and a centrum that is wider than longer (Fig. 4P). However, the sacral transverse processes of *Beelzebufo* are slightly more expanded distally than that preserved for *Cretadhefdaa* (Fig. 4R).

Another neobatrachian from Gondwana with an ornamented skull is *Baurubatrachus pricei Báez & Perí, 1989* from the Crato Formation of Brazil (Upper Early Cretaceous). The poor preservation of the frontoparietals of the holotype (and only known specimen), which is still embedded in matrix, prevents comparisons of the braincase of *Cretadhefdaa* to

*Baurubatrachus*. However, its frontoparietals seem to be similar in having ornamentation comprised of pits and ridges that extend posteriorly to the margin of the foramen magnum. *Cretadhefdaa* also differs from *B. pricei* in having a fully ossified dorsal margin of the foramen magnum, and a foramen for the arteria orbitonasalis dorsal to the prootic foramen. The maxilla of *Cretadhefdaa* is similar to *B. pricei* in having ornamentation on the lateral surface of the pars facialis that extends ventrally to the pars dentalis, but differs in lacking a distinct palatine shelf. *Cretadhefdaa* differs from *B. pricei* in having an occipital flange and a system of recesses on the posterodorsal region of the braincase. *Cretadhefdaa* also differs from *B. pricei* in having more slender and shorter neural spines on presacral vertebrae and slightly expanded sacral transverse processes.

In his 2012 review, Agnolin described several specimens as *Calyptocephalella satan*, the oldest calyptocephalellid described (*Agnolin, 2012*). Although these specimens need to be reassessed (*Báez & Gómez, 2018*) and likely represent more than one taxon (*Muzzopappa et al., 2020*), their attribution to Neobatrachia is certain. *Cretadhefdaa* resembles *C. satan* in having dermal skull bones covered with an ornamentation of pits and ridges, but differs in lacking a distinct palatine shelf (all calyptocephalellids exhibit a distinct palatine shelf; *Muzzopappa & Báez, 2009*; *Agnolin, 2012*), in having fused frontoparietals without a median suture, and in having an occipital flange on the frontoparietals (Fig. 1A). The postcranial elements of *Cretadhefdaa* resemble *C. satan* in having procoelous vertebrae with anteroposteriorly elongate centra for the anterior presacral vertebrae, and shorter centra for posterior presacral and sacral vertebrae (*Agnolin, 2012*). The sacral vertebra bears a bicondylar articulation in both taxa, but *Cretadhefdaa* differs in having sacral transverse processes that are weakly expanded distally, whereas *C. satan* exhibits greatly expanded sacral transverse processes (*Agnolin, 2012*: figs. 10A and 10B).

One last ornamented Cretaceous neobatrachian taxon is *Hungarobatrachus szukacsi* from the Late Cretaceous of Hungary. Its vertebral elements are not known, but several skull fragments were recently described (*Venczel, Szentesi & Gardner, 2021*). Both taxa have fused frontoparietals without a trace of suture along their medial margin. However, *Cretadhefdaa* differs from *H. szukacsi* in having a system of recesses on each side of the posterior surface of its frontoparietals (divided by the foramen magnum) with the foramen for the occipital artery opening in a deep recess and an occipital flange on the frontoparietals. In *H. szukacsi*, the posterior surface of the frontoparietals is smooth with a slight depression and the foramen for the occipital artery opens on each side of the foramen magnum (*Venczel, Szentesi & Gardner, 2021*: fig. 3). The frontoparietals of *H. szukacsi* also bear an incrassatio frontoparietalis on the ventral surface whereas *Cretadhefdaa* does not. The maxilla of *Cretadhefdaa* differs from that of *H. szukacsi* in lacking a distinct palatine shelf (*Venczel, Szentesi & Gardner, 2021*: fig. 5).

## Comparison to hyperossified extinct ranoids

Two other hyperossified taxa are relevant for comparisons to *Cretadhefdaa*: *Rocekophryne ornata* (*Rage et al., 2021*) from the Early Eocene of Algeria (*Rage et al., 2021*) and *Thaumastosaurus servatus* from the Middle to Late Eocene of southwestern France (*Lemierre*

*et al., 2021*). These are the oldest occurrences of ornamented ranoids in the fossil record (*Lemierre et al., 2021*; *Rage et al., 2021*).

*Rocekophryne ornata* is known from fragmentary cranial and postcranial remains. *Cretadhefdaa* resembles *Rocekophryne* in having fused frontoparietals without a median suture and bearing an ornamentation of pits and ridges, an occipital flange, and in lacking an incrassatio frontoparietalis on the ventral surface of the frontoparietals. In addition, *Cretadhefdaa* and *Rocekophryne* both bear ornamentation on the lateral surface of the pars facialis of the maxilla that extends ventrally to the pars dentalis (Fig. 3F). However, *Cretadhefdaa* differs in lacking a lateral flange on the posterior surface of the frontoparietal, lacking a distinct palatine shelf, and in having very short paraoccipital processes (well-developed in *Rocekophryne*; *Rage et al., 2021*: figs. 3A–3F) and a series of recesses on the posterodorsal surface of the braincase. In addition, the sacral vertebra of *Rocekophryne* bears an anterior condyle (instead of an anterior cotyle in *Cretadhefdaa*) that indicates that the vertebral column is diplasiocoelous (*Rage et al., 2021*: figs. 4A and 4B) and possesses transverse processes that are circular in lateral view (not circular in *Cretadhefdaa*).

*Thaumastosaurus servatus* is known from fragmentary remains and three partially complete and articulated skeletons (*Rage & Roček, 2007*; *Lemierre et al., 2021*). As with *R. ornata*, *Cretadhefdaa* and *T. servatus* have fused and ornamented frontoparietals without a medial suture. The anterior surface of the prooticooccipitals of both taxa exhibit a well-delimited but shallow and narrow groove for the jugular vein (*Rage & Roček, 2007*: fig. 7; *Lemierre et al., 2021*: fig. 8F). However, *Cretadhefdaa differs* from *T. servatus* in having an occipital flange and reduced paraoccipital processes, lateromedially compressed occipital condyles (instead of crescent shaped), and a series of recesses in the posterodorsal surface of the braincase (Fig. 1). *Cretadhefdaa* also differs from *T. servatus* in lacking a single, tapered posterior process of the parasphenoid and an incrassatio frontoparietalis on the ventral surface of the frontoparietals (Fig. 1C). In addition, the vertebral column of *T. servatus* is diplasiocoelous instead of procoelous as in *Cretadhefdaa*.

## Comparisons to extant hyperossified hyloids

*Cretadhefdaa* shares numerous characters with ornamented extant Neobatrachia. Most of these similarities are associated with hyperossification, but two characters deserve further attention. The first is the presence of contact between the squamosal and frontoparietals, which occurs frequently (but not uniquely) in Hyloidea (*e.g.*, Calyptocephalellidae, Ceratophryidae, or the hylid *Triprion* Cope, 1866). The second is the series of recesses on the posterodorsal surface of the braincase in *Cretadhefdaa*. This is known only in *Beelzebufo* and in *Ceratophrys* Wied-Neuwied, 1824 (*Evans et al., 2014*; Figs. 5B and 5C). However, both *Cretadhefdaa* and *Beelzebufo* differ from *Ceratophrys* in having the foramen for the occipital artery located in the central recess, whereas it is found in the medial recess in extant taxa (Fig. 5). The braincase of *Ceratophrys* is similar to *Cretadhefdaa* in having fused frontoparietals, no distinct posterior process, and barely distinct paraoccipital processes. *Cretadhefdaa* differs from *Ceratophrys* in having an occipital flange, a well-delimited groove for the jugular vein, and in lacking the expanded "table" atop the neural spine of presacral vertebrae. The extant *Triprion* differs from *Cretadhefdaa* in having a frontoparietal

extending posteriorly up to the end of the epiotic eminence, covering it dorsally. *Triprion petasatus* Cope, 1865 also lacks the system of recesses on the posterodorsal surface of the braincase. *Triprion spatulatus* Günther, 1882 bears recesses on it posterodorsal region of the braincase, but differs from *Cretadhefdaa* in having the foramen for the arteria occipitalis not located within a recess.

*Neobatrachia?* Reig, 1958
*Ranoidea?* Rafinesque, 1814

### Forelimb (UCRC-PV104)

This specimen is an incomplete humerus missing its proximal end and part of the diaphysis (Fig. 6). The diaphysis is straight, and a thin ventral ridge on the proximal end of the bone extends distally to the midlength of the diaphysis (Figs. 6A and 6C). The fossa cubitalis is very reduced, being shallow and not well-delimited, and visible in ventral view only as a thin crescent around the humeral ball (Fig. 6A). The humeral ball is large and in-line with the main axis of the diaphysis. The epicondyles are not symmetrical, with the ulnar epicondyle well-developed and the radial epicondyle reduced and barely visible in ventral view (Fig. 6A). In dorsal view, the olecranon scar is short, with a tapered and pointed end (Fig. 6B).

### Comparisons

The combination of a large humeral ball and asymmetrically developed epicondyles is diagnostic for most Neobatrachia (*Prasad & Rage, 2004*; *Rage, Pickford & Senut, 2013*), although this combination of characters has not been evaluated in phylogenetic analyses. The presence of a straight diaphysis, a humeral ball in line with the axis of the diaphysis, and a shallow, poorly delimited fossa cubitalis are found in most ranoids (*Rage, Pickford & Senut, 2013*; *De Lapparent de Broin et al., 2020*). It differs from the humerus of *Thaumastosaurus servatus*, one of the earliest known ranoids, in having a crescent-shaped fossa cubitalis (triangular in *T. servatus*) and a less developed ulnar epicondyle. Among the Cretaceous neobatrachian taxa, only *Eurycephalella alcinae* (*Báez, Moura & Gómez, 2009*) and *Arariphrynus placidoi* (*Leal & Brito, 2006*) have preserved humeri with their ventral surface exposed. The humerus of *A. placidoi* differs from UCRC-PV104 in having two well-developed epicondyles (instead of a reduced radial epicondyle) and a deep fossa cubitalis (instead of a shallow fossa in UCRC-PV104).

These comparisons suggest that UCRC-PV104 should be referred to the Neobatrachia. UCRC-PV104 shares several characters with extant and extinct Ranoidea, as well as with the oldest (putative) member of the Ranoidea (*Thaumastosaurus servatus*). However, because no phylogenetic analyses have yet shown synapomorphies for Ranoides related to the humerus, we refer this fossil to the Neobatrachia and recognize the assignment to Ranoidea as tentative.

Incertae Sedis

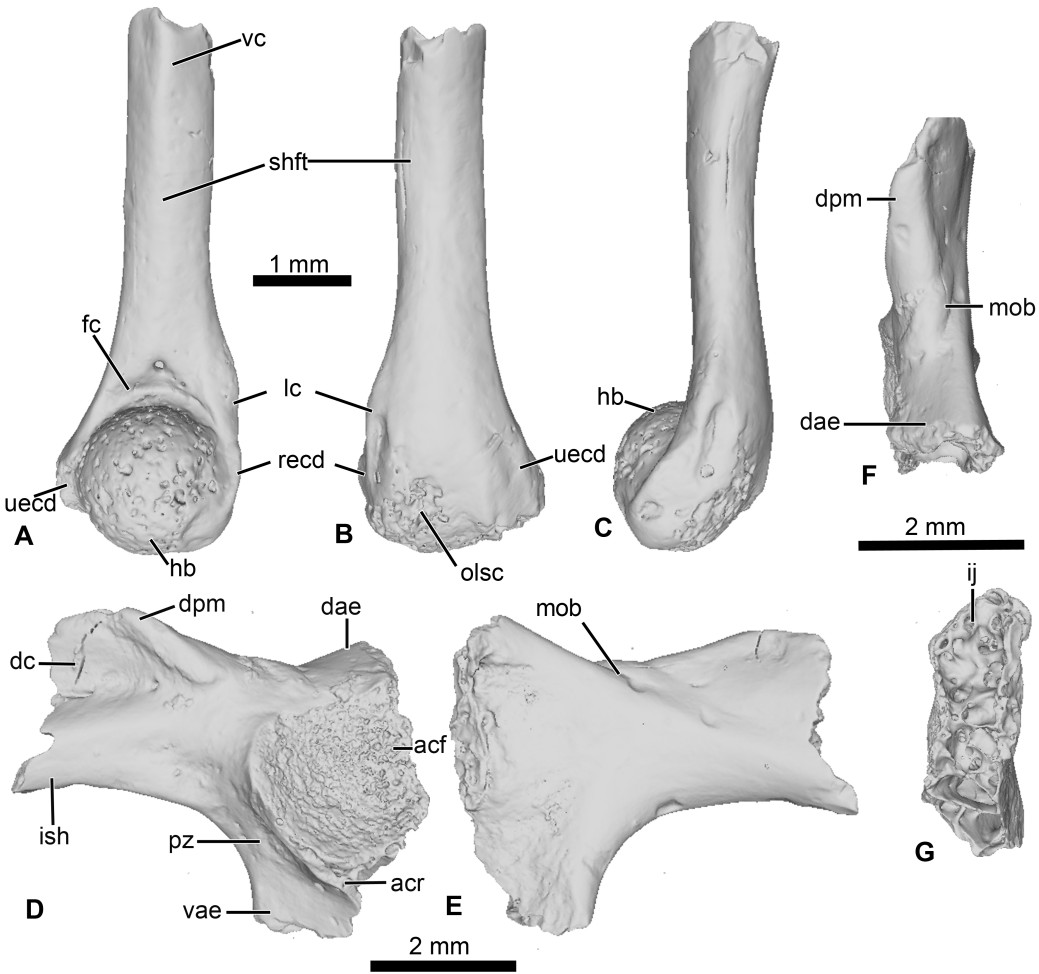

**Figure 6   Neobatrachia? and indeterminate ilium from Kem Kem beds.** (A–C) UCRC-PV104, incomplete humerus in (A) ventral; (B) dorsal and (C) lateral views; (D–G) UCRC-PV105, left ilium in (D) lateral; (E) medial; (F) posterior and (G) dorsal views. Abbreviations: acf, acetabular fossa; acr, acetabular rim; dae, dorsal acetabular expansion; dc, dorsal crest; dpm, dorsal prominence; fc, fossa cubitalis; hb, humeral ball; ij, ilioischiatic juncture; ish, iliac shaft; lc, lateral crest; mob, medial oblique ridge; olsc; olecranon scar; pz, preacetabular zone; recd, radial (lateral) epicondyle; shft, shaft; uecd, ulnare (medial) epicondyle; vae, ventral acetabular expansion; vc, ventral crest.

## Pelvic girdle (UCRC-PV105)

This element is an incomplete left ilium, preserving most of its acetabular region. UCRC-PV105 bears a high and well-developed dorsal crest, although its extension on the iliac shaft is unknown (Figs. 7I–7J). The dorsal crest appears to be lacking its dorsalmost portion, indicating that it was more extensive (Figs. 7I and 7K). The dorsal prominence is low and elongate anteroposteriorly, and the dorsal protuberance is strongly oriented laterally (Fig. 7K). The acetabular rim is well developed on its ventral region. Although not complete, both the dorsal and ventral acetabular expansions are developed. The dorsal acetabular expansion is inclined posteromedially (Fig. 7I). The ventral acetabular expansion is poorly preserved. However, the preserved portion shows it was well-developed (Fig. 4I). The

preacetabular angle is obtuse and the preacetabular zone is narrow (Fig. 7I). In medial view, a shallow but well delimited medial ridge is present, starting from the base or the dorsal acetabular expansion to the anteriormost preserved portion (Fig. 7J). In posterior view, the ilioischiadic juncture is moderately wide and an interiliac tubercle is absent (Fig. 7L).

## Comparisons

Ilia are one of the most common anuran elements recovered in the fossil record (*Roček, 2000*; *Rage & Roček, 2003*; *Roček, 2013*; *Roček et al., 2013*; *Gardner & Rage, 2016*) and several authors have proposed characters to identify the ilia of the different clades (*Gardner et al., 2010*; *Gómez & Turazzini, 2016*; *Matthews, Keeffe & Blackburn, 2019*). However, these are largely based on extant anurans and can be difficult to apply to Mesozoic anurans (*Roček et al., 2010*; *Roček, 2013*). The presence of a well-developed dorsal crest is found in several clades (Alytoidea, Pipoidea, and Neobatrachia, especially Ranoidea), but likely reflects similarity in locomotion rather than close phylogenetic relationships (*Roček, 2013*). The absence of an interiliac tubercle is diagnostic for many neobatrachians, with notable exceptions such as *H. szukacsi* and the aquatic hylid *Pseudis* Wagler 1830 (*Gómez & Turazzini, 2016*; *Venczel, Szentesi & Gardner, 2021*). However, the utility of this character has not been tested thoroughly in a taxon-rich phylogenetic analysis (*Gómez & Turazzini, 2016*). *Agnolin (2012)* argued that the presence of a broad preacetabular zone and large acetabular fossa was diagnostic for the Calyptocephalellidae but this was not evaluated in a phylogenetic analysis and may represent an example of convergent evolution. There are no characters that allow for a precise attribution of this ilium (UCRC-PV105) to the other anurans from the Kem Kem or other specific anuran lineages.

## Phylogenetic analyses

Recent phylogenetic analyses (*Báez & Gómez, 2018*; *Lemierre et al., 2021*) are based on a similar dataset. This dataset was first elaborated by *Báez, Moura & Gómez (2009)*, based on the dataset of *Fabrezi (2006)* that was developed for a phylogenetic analysis of ceratophryids. The dataset from *Báez, Moura & Gómez (2009)* includes 42 taxa—three of which are extinct taxa—and 75 characters. In a separate analysis, *Báez & Gómez (2018)* modified the dataset from *Fabrezi (2006)* further by adding 29 neobatrachian taxa and redefining some characters to test the impact of characters related to hyperossification. They expanded the taxon sampling to 71 taxa and added 68 characters (for a total of 143 characters), as well as redefined several characters. Finally, *Lemierre et al. (2021)* further enlarged the dataset from *Báez & Gómez (2018)*, by adding 15 extant natatanuran ranoid taxa (for a total of 20 natatanuran taxa). The vast majority of extant anurans belong to the Neobatrachia (*Feng et al., 2017*), which includes two large clades, Hyloidea and the Ranoidea. To date, phylogenetic analyses based solely on morphological characters (*e.g.*, *Scott, 2005*) do not recover many of the clades found in recent molecular phylogenetic analyses (*e.g.*, *Roelants et al., 2007*; *Feng et al., 2017*; *Jetz & Pyron, 2018*; *Hime et al., 2021*). To evaluate the phylogenetic placement of *Cretadhefdaa*, we analyzed our character matrix using different sets of assumptions as well as one analysis using a constraint tree reflecting recent results from molecular phylogenetic analyses.

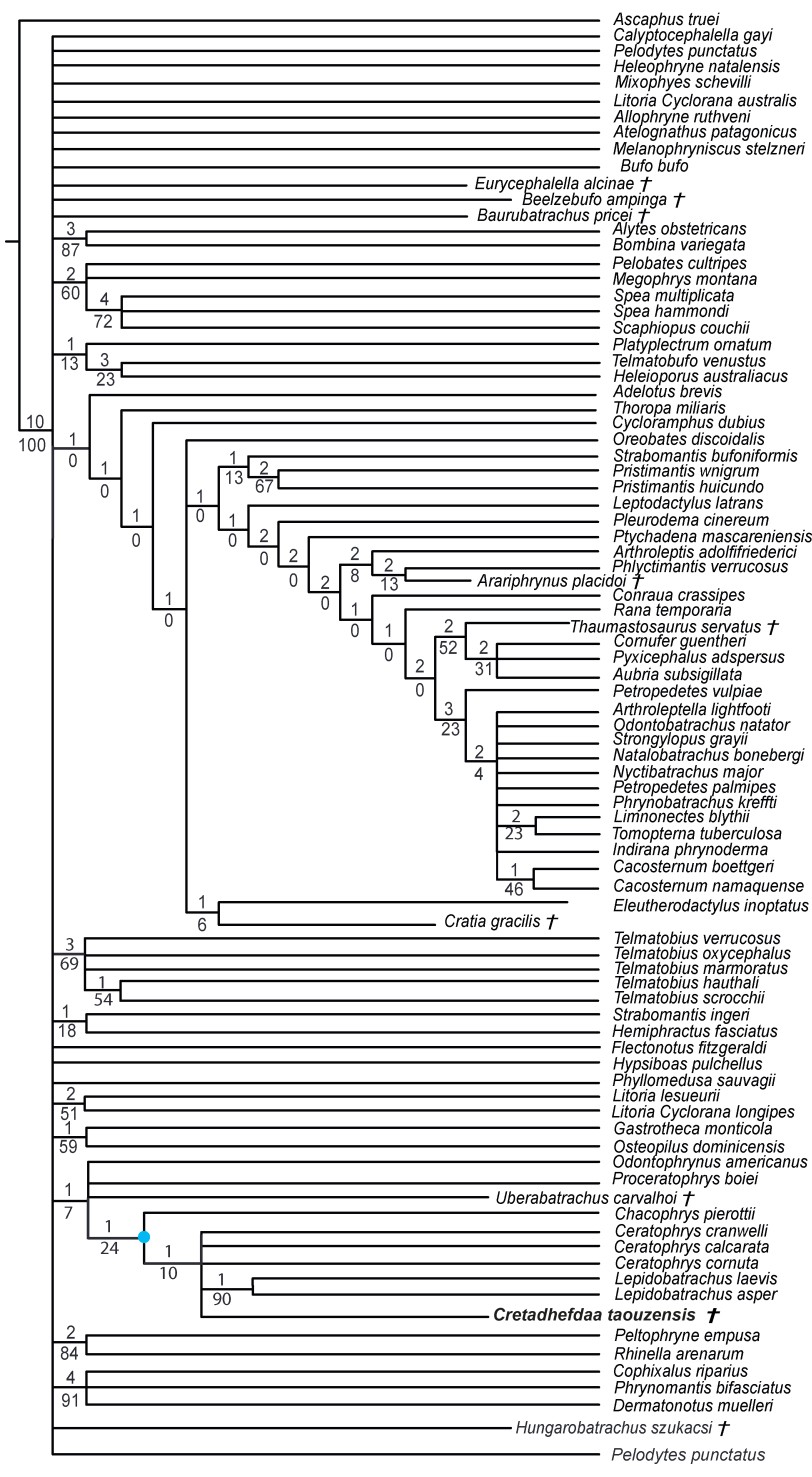

**Figure 7 Strict consensus of 60 MPTs of 1362 steps (CI = 0.139; RI = 0.418) from the analysis under EW.** † represents extinct taxon, light blue circle represents Ceratophryidae node; numbers above branches designate Bremer support; those below are bootstrap frequencies.

## RESULTS

We obtained 60 MPTs (most parsimonious trees) of 1362 steps (CI = 0.139; RI = 0.418) with the analysis performed under equal weight with cline characters ordered. The strict consensus (Fig. 7) shows large polytomies, and the monophyly of the Neobatrachia is not recovered. This seems related to the uncertainties regarding the position of *Arariphrynus placidoi*, and the lack of characters scored for *Cretadhefdaa* and *Hungarobatrachus szukacsi* (13 and 11% of characters scored, respectively). *Cretadhefdaa* is recovered within a clade containing *Uberabatrachus carvalhoi* Baéz et al. 2012 and the Ceratophryidae. This clade is supported by three synapomorphies, all of which are character states found in other groups of frogs: (1) a position of articulation of lower jaw and skull at the level of occiput (character 61: 0 >1); (2) cotyle of the atlas widely separated (76: 1 >2) and (3) angle between iliac shaft and ventral acetabular expansion obtuse (125: 1 >2). *Cretadhefdaa* is placed within this clade in a polytomy with the Ceratophryidae. This clade is supported by five synapomorphies mainly related to hyperossified cranial characters (see Appendix S4).

When excluding *Arariphrynus*, we obtained 10 trees of 1,355 steps. The strict consensus (CI = 0.174; RI = 0.556; Fig. 8) shows a trichotomy with Pelobatoidea, *Heleophryne* Sclater 1898, and the remaining Neobatrachia. The 'Neobatrachia' (the clade exclusive of *Heleophryne*) is supported by a five synapomorphies: (1) otic ramus of the squamosal short, overlapping only the most lateral portion of the crista parotica (9: 0 >1); (2) absence of process or crest on the anterior margin of the scapula (114: 3 >0); (3) configuration of the postaxial carpals as ulnare free, 3+4+5 (119: 0 >2); (4) well developed posterodorsal expansion of the ischium (131: 0 >1) and (5) horizontal pupil shape (143: 0 >2). Among the Neobatrachia, we recovered a large hyperossified clade, supported by six synapomorphies (see Appendix S4). *Hungarobatrachus* is within a poorly supported trichotomy with *Eurycephalella* and *Calyptocephalella* Strand 1928, for which there are three synapomorphies: (1) contact between lamella alaris of the squamosal and frontoparietals on the dorsal surface of the otic capsule (8: 0 >2); (2) anterior ramus of the pterygoid not reaching planum antorbitale (12: 0 >1) and (3) postaxial carpal with ulnare and 3 free (119: 2 >1). *Cretadhefdaa* is recovered within a large polytomy with extant Ceratophryidae, poorly supported by four synapomorphies (see Appendix S4).

In analyses using a topological constraint (and excluding *Arariphrynus placidoi*), we obtained 190 trees, with a score of 1,395 steps. The strict consensus (CI = 0. 126, RI = 0. 247; Fig. 9) shows a monophyletic Neobatrachia, Ranoidea, and Hyloidea, but all of the monophyly of each was enforced in the constraint tree. Within Hyloidea, most taxa are placed within a large unresolved clade (Fig. 9). *Cretadhefdaa* is recovered in a large polytomy within Hyloidea as are *Baurubatrachus*, *Beelzebufo*, *Cratia* (*Báez, Moura & Gómez, 2009*), *Eurycephalella*, *Hungarobatrachus*, and *Uberabatrachus*. The only extinct taxon to be recovered elsewhere in the phylogeny is *Thaumastosaurus*, which is recovered in a clade of Ranoidea with *Aubria* Boulenger 1917, *Cornufer* Tschudi 1838, and *Pyxicephalus* Tschudi 1838.

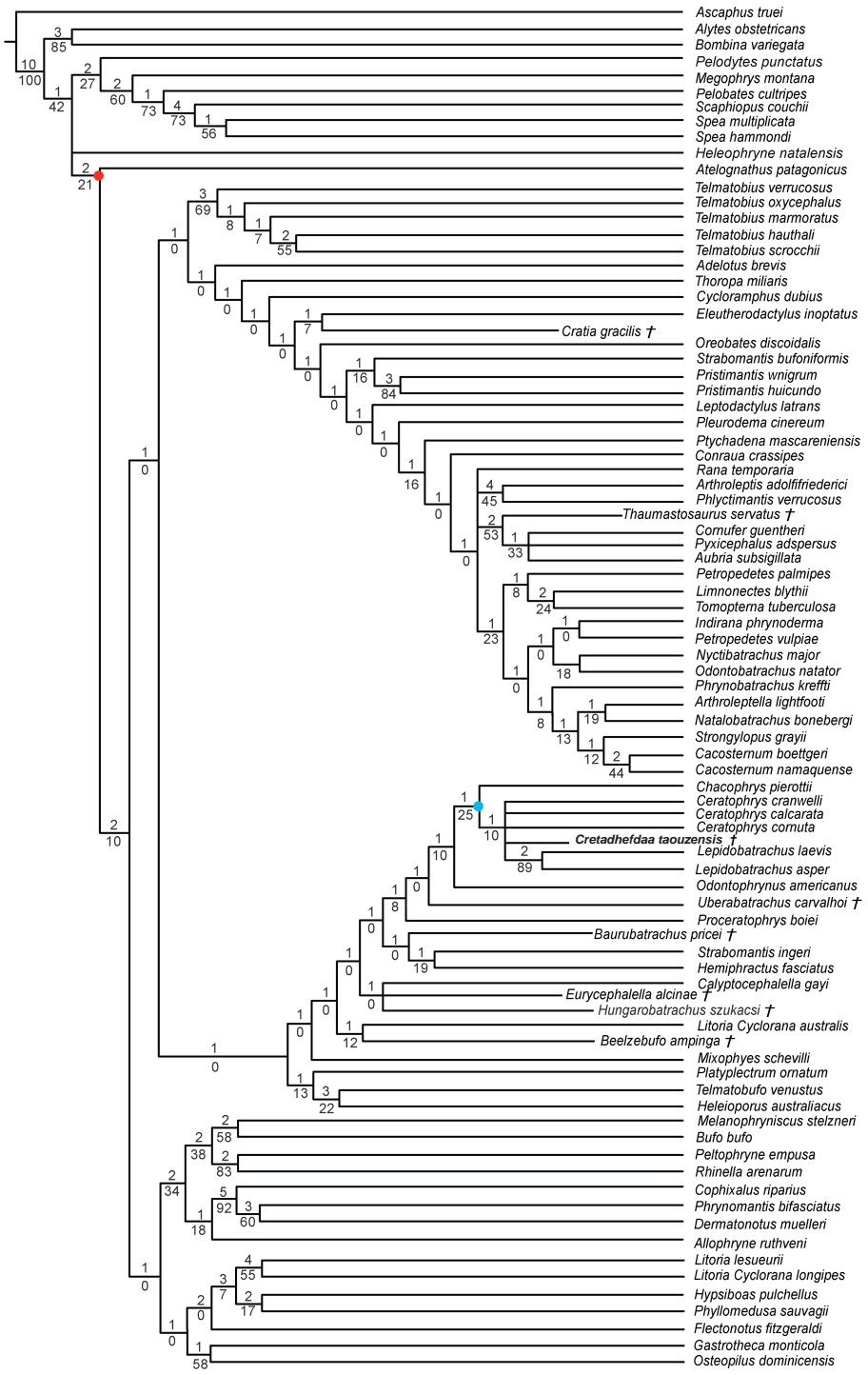

**Figure 8** **Strict consensus of 10 MPTs of 1355 steps (CI = 0.174; RI = 0.556) from the analysis under EW excluding *Arariphrynus placidoi*.** † represents extinct taxon, red circle represents Neobatrachia node (excluding *Heleophryne*); numbers above branches designate Bremer support; those below are bootstrap frequencies and light blue circle represents Ceratophryidae node.

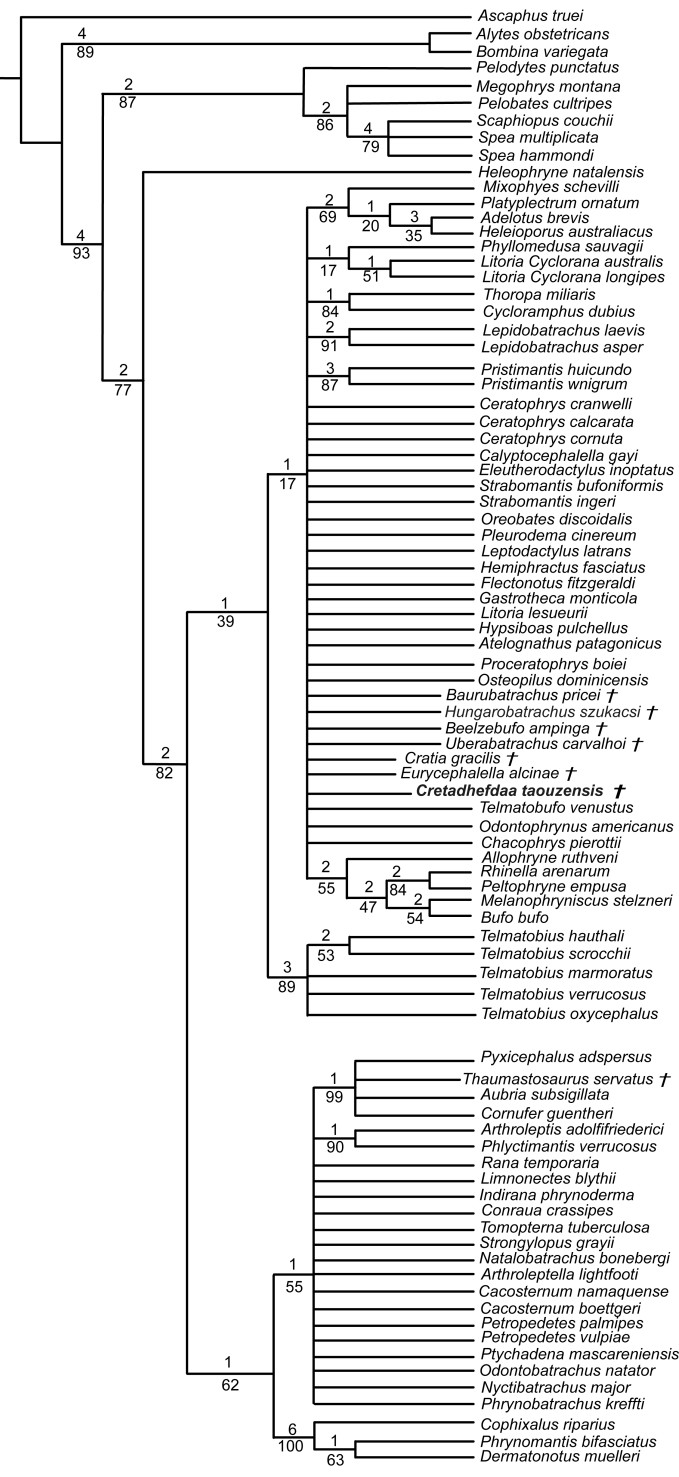

**Figure 9  Strict consensus of 190 MPTs of 1395 steps (CI = 0.126; RI = 0.247) from the analysis under EW, excluding *Arariphrynus placidoi* and using a constraint topology based on molecular phylogenetic analyses.** † represents extinct taxon and numbers above branches designate Bremer support; those below are bootstrap frequencies.

## DISCUSSION

### Phylogenetic analyses

The poor resolution of the topology obtained when performing phylogenetic analysis under equal weights is not surprising. *Hungarobatrachus szukacsi* has only 16 scored characters within the dataset, none of which are clear neobatrachian synapomorphies, and the skeleton of *Arariphrynus* is very incomplete leading to few scored characters, especially those for the pectoral girdle and vertebrae (51 scored characters in total; see *Báez, Moura & Gómez, 2009*). In addition, most of the scored cranial characters for *Hungarobatrachus* and *Cretadhefdaa* are linked to hyperossification, a recurrent feature in anuran evolution (see above) that likely obscures the phylogenetic relationships of *Cretadhefdaa*.

The phylogenetic positions of *Cretadhefdaa* and *Hungarobatrachus* are similar to several hyperossified extinct Cretaceous taxa by being close to either the Ceratophryidae or Calyptocephalellidae. Recent analyses (*Báez & Gómez, 2018*) have highlighted that convergence due to hyperossification likely plays a role in the position recovered for other hyperossified extinct neobatrachian taxa. This could influence the position of *Cretadhefdaa* as well. Nevertheless, the combination of characters of *Cretadhefdaa* confirms its assignment to Neobatrachia. In addition, one character mentioned in the description of the braincase, the presence of a series of recesses in posterodorsal region of the braincase, deserves attention. In addition to *Cretadhefdaa*, a similar (but not clearly homologous) morphology has only been identified in *Beelzebufo* and in the Ceratophryidae (except in *Chacophrys* Reig and Limeses 1963). To our knowledge, this character has not been used in phylogenetic analyses (*e.g.*, *Gómez & Turazzini, 2021*). However, the two extant taxa possessing these recesses are closely related (*Ceratophrys* and *Lepidobatrachus* Budgett, 1899), and the extinct *Beelzebufo* has been proposed as a stem member of the Ceratophryidae (*Báez & Gómez, 2018*; *Lemierre et al., 2021*). Interestingly, *Cretadhefdaa* is recovered in a more crownward position within Ceratophryidae than *Beelzebufo*, even in other analyses (*Báez & Gómez, 2018*; *Lemierre et al., 2021*). It is necessary to test the phylogenetic significance of this character to confirm this hypothesis, which is beyond the scope of this paper. When using a topological constraint based on recent phylogenomic analyses, most extinct taxa—including *Cretadhefdaa*—included in the analysis were recovered as part of Hyloidea, though as part of a large polytomy. In conclusion, our phylogenetic analyses point to *Cretadhefdaa* being within the Neobatrachia, even if most of the synapomorphies diagnostic of this clade are not scored, and several analyses support a hyloid affinity.

### Paleobiogeographical implications

Neobatrachians are known in the fossil record during the Late Cretaceous from three main locations: Madagascar (Maastrichtian; *Evans et al., 2014*), Europe (Campanian; *Venczel, Szentesi & Gardner, 2021*), and South America (Maastrichtian; *Báez & Gómez, 2018*). The South American fossil record is of particular importance with numerous taxa known from articulated specimens (*Báez, Moura & Gómez, 2009*; *Báez & Gómez, 2018*; *Agnolin et al., 2020*; *Moura et al., 2021*). In contrast, only fragmentary remains of two taxa have been recovered from Madagascar and Europe (*Evans, Jones & Krause, 2008*; *Evans et al., 2014*; *Venczel, Szentesi & Gardner, 2021*). There are other reports of neobatrachians from the

Cretaceous (*Báez & Werner, 1996*; *Prasad & Rage, 2004*; *Rage, 1984*; *Rage et al., 2020*) but the attribution of these to the Neobatrachia remains uncertain because diagnostic elements are often not preserved and these other fossils have not been included in phylogenetic analyses. Because *Cretadhefdaa* is from the Mid-Cenomanian, it is the oldest neobatrachian of Africa.

The oldest occurrence of the Neobatrachia is from the Brazilian Crato Formation (*Leal & Brito, 2006*; *Báez, Moura & Gómez, 2009*; *Agnolin et al., 2020*; *Moura et al., 2021*), which preserves extinct anurans from the Aptian (Early Cretaceous). However, *Cretadhefdaa* is still the oldest occurrence of Neobatrachia outside of South America. The Neobatrachia began to diversify during the earliest Cretaceous, including an early split into two major lineages, Hyloidea and Ranoidea, each of which was largely restricted to a portion of western Gondwana, respectively, South America and Africa (*Frazão, Da Silva & Russo, 2015*; *Feng et al., 2017*). Time-calibrated molecular phylogenetic analyses (*e.g.*, *Feng et al., 2017*) suggest that by 96–95 Ma (*i.e.*, the period from which *Cretadhefdaa* was recovered), the Neobatrachia was already separated into a number of lineages that are restricted today to specific biogeographic regions. These include the Myobatrachidae of Australia, the hyloids of South America, the Microhylidae (widespread today across the tropics), the Afrobatrachia of sub-Saharan Africa, the natatanuran ranoids, and the lineage leading to the Sooglossidae and Nasikabatrachidae that are today restricted, respectively, to the Seychelles Islands and the Western Ghats of India. There remains ample opportunity for both additional sampling and study of neobatrachian fossils from Gondwanan landmasses that could add new insights into the early evolution and biogeography of these major extant frog lineages that diversified in the Early Cretaceous.

The current absence of Ranoidea from the Cretaceous fossil record is puzzling. Except for undescribed and unillustrated material that was attributed to Ranoidea two decades ago (*Báez & Werner, 1996*), there are surprisingly few ranoid fossils especially in comparison to the hyloid fossils discovered in South America, Europe, and Africa. Their absence could be due to several factors. The first and most obvious is the lack of anuran specimens from the fossil record of Africa, due both to a lack of targeted collecting and little academic research on existing material. One example that highlights this problem is the Pyxicephalidae, a clade of ranoids endemic to Africa (*Channing & Rödel, 2019*) and for which time-calibrated molecular phylogenetic analyses suggest a divergence from other natatanurans around 60 Ma (Early Palaeocene). Yet, the oldest occurrence of this family is *Thaumastosaurus* from the Middle-Late Eocene of Europe, whereas the earliest African fossil is from only 5 Ma (*Matthews et al., 2015*; *Lemierre et al., 2021*). The large gap in the fossil record of this family is found in many other families of Ranoidea, and many clades with an African origin completely lack a fossil record. Another bias could be that the vast majority of Ranoidea are not hyperossified anurans, including many small-sized species, and thus less likely to be preserved as intact and diagnosable fossils. In addition, numerous synapomorphies of Ranoidea are for postcranial elements, such as the vertebrae and the pectoral girdle, that are less likely to be identified and/or preserved (*Scott, 2005*; *Frost et al., 2006*). A final bias is simply that there has been sustained interest from South American paleontologists in

the fossil record of anurans from countries such as Bolivia, Brazil, and Argentina, whereas there have been exceedingly few African paleontologists dedicated to studying anurans.

## CONCLUSION

Our study confirms the report of *Rage & Dutheil (2008)* that at least three anuran taxa are present in the Kem Kem beds of Morocco. The newly described *Cretadhefdaa taouzensis* can be attributed to the Neobatrachia, making it both the oldest occurrence of the clade outside of South America and only the second occurrence in the Cretaceous of Africa. Several postcranial bones also point to an affinity with the Neobatrachia but cannot be associated definitively with either *Cretadhefdaa* or another taxon. The presence of a neobatrachian in the Kem Kem in the Cenomanian demonstrates that neobatrachians were already widespread on Gondwana during the earliest Late Cretaceous.

**Institutional abbreviations**

| | |
|---|---|
| **MNHN** | Muséum National d'Histoire Naturelle, Paris (France) |
| **UCRC-PV** | University of Chicago research collection, Chicago (USA) |

## ACKNOWLEDGEMENTS

We thank Paul Sereno (University of Chicago) for access to and loan of the specimens, and Edward Stanley (University of Florida) for CT-scanning them. Processing of tomographic data was undertaken at the 3D imaging facilities Lab of the UMR 7207 CR2P (MNHN CNRS UPMC, Paris). We thank María Vallejo-Pareja for comments on a draft of this manuscript and both Zbyněk Roček and David Cannatella for constructive criticism.

### Funding

Funding for CT-scanning of both fossils and comparative material was supported by the US National Science Foundation (DBI-1701714 to David C. Blackburn). Funding for study of materials was supported by a grant from the Fondation pour la Recherche sur la Biodiversité (FRB, France) to Alfred Lemierre. The funders had no role in study design, data collection and analysis, decision to publish, or preparation of the manuscript.

### Grant Disclosures

The following grant information was disclosed by the authors:
US National Science Foundation: DBI-1701714.
Fondation pour la Recherche sur la Biodiversité.

### Competing Interests

The authors declare there are no competing interests.

## Author Contributions

- Alfred Lemierre conceived and designed the experiments, performed the experiments, analyzed the data, prepared figures and/or tables, authored or reviewed drafts of the article, and approved the final draft.
- David C. Blackburn conceived and designed the experiments, authored or reviewed drafts of the article, and approved the final draft.

## Data Availability

The phylogenetic dataset is available in the Supplemental File.

The CT-scan and 3D models are available on MorphoSource: Project 000426961. https://www.morphosource.org/projects/000426961?locale=en

- UCRC-PV94, Incomplete posterior braincase, Cretadhefdaa taouzensis, CT Scan, https://doi.org/10.17602/M2/M168041; 3D Model, https://doi.org/10.17602/M2/M427199.

- UCRC-PV94_Inner ear, Endocast of Inner Ear, Cretadhefdaa taouzensis, CT Scan, https://doi.org/10.17602/M2/M168041; 3D Model, https://doi.org/10.17602/M2/M427199.

- UCRC-PV95, Incomplete squamosal, Cretadhefdaa taouzensis, CT Scan, https://doi.org/10.17602/M2/M351726; 3D Model, https://doi.org/10.17602/M2/M427218.

- UCRC-PV96, Incomplete maxilla, Cretadhefdaa taouzensis, CT Scan, https://doi.org/10.17602/M2/M389771; 3D Model, https://doi.org/10.17602/M2/M427196.

- UCRC-PV97, presacral vertebra, Cretadhefdaa taouzensis, CT Scan, https://doi.org/10.17602/M2/M351731; 3D Model, https://doi.org/10.17602/M2/M427215.

- UCRC-PV98, presacral vertebra, Cretadhefdaa taouzensis, CT Scan, https://doi.org/10.17602/M2/M351736; 3D Model, https://doi.org/10.17602/M2/M427212.

- UCRC-PV101, presacral vertebra, Cretadhefdaa taouzensis, CT Scan, https://doi.org/10.17602/M2/M351753; 3D Model, https://doi.org/10.17602/M2/M427227.

- UCRC-PV103, Sacral vertebra, Cretadhefdaa taouzensis, CT Scan, https://doi.org/10.17602/M2/M351822; 3D Model, https://doi.org/10.17602/M2/M427202.

- UCRC-PV104, Humerus, Neobatrachia ?, CT Scan, https://doi.org/10.17602/M2/M351812; 3D Model, https://doi.org/10.17602/M2/M427209.

- UCRC-PV105, Ilium, Anura Indet., CT Scan, https://doi.org/10.17602/M2/M351817; 3D Model, https://doi.org/10.17602/M2/M427206.

## New Species Registration

The following information was supplied regarding the registration of a newly described species:

Publication LSID: urn:lsid:zoobank.org:pub:DCACD333-53AA-4A6D-A0F0-9F9C180F0DDC

Genus name: Cretadhefdaa: urn:lsid:zoobank.org:act:F144A555-3232-4FA0-8405-217E3DA55331

Species name: Cretadhefdaa taouzensis: urn:lsid:zoobank.org:act:A16F9829-615A-4405-AEBF-E62785E1BB7D

## Supplemental Information

Supplemental information for this article can be found online at http://dx.doi.org/10.7717/peerj.13699#supplemental-information.

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
