# Peer review of "A new genus and species of frog from the Kem Kem (Morocco), the second neobatrachian from Cretaceous Africa"

_PeerJ, doi:10.7717/peerj.13699_

## Round 0.1 · original submission · Major Revisions

I have now received two very detailed reviews on your manuscript. While both of them saw merit in your work, they also pointed out a number of aspects that deserve author's attention before a final decision is made. Notice R2 asks for more details in many instances of the Methods. Pay clser attention to the synapomorphy highlighted by R2, bicondylar sacrum, which appears to be a diagnostic character shared by Neobatrachia and confirms the position of the new taxon on the tree. Both reviewers also points to some issues with labelling in Fig 1 and 2. Notice R1 questions the reliability of the six characters used as diagnostic to separate this taxon from others. The same reviews makes many detailed recommendations on how to standardize terminology (confusing intermix of Latin and English) and also the description of many other characters. Take a closer look at his review.

·

Excellent Review

This review has been rated excellent by staff (in the top 15% of reviews)
EDITOR COMMENT
This review was particularly helpfull because it pointed out several aspects of the morphological description that need to be improved, with consequences for the diagnostic characters used to differentiate the new taxon from others. In addition, the reviewer also catched several instances in which terminology used was confusing. If authors follow his comments, the paper will certainly be improved

Basic reporting

- Some minor grammatical errors can be found (e.g. anuran remains, instead of “anurans remains”; Abstract, line 8). However, I am not a native English speaker, so I cannot assess this aspect. This is why I recommand linguistic editing (e.g., lines 56-58: “that cannot be easily incorporated”, instead of “remains ... that are not easily incorporated into phylogenetic analyses”, and many others).
- Introduction is informative, but Material and Methods should be completed by information which anatomical terminology was followed. I strongly recommand that the authors use standard anatomical terminology (e.g., from Sanchiz 1998), because some of their terms (especially those which are used in anglicized form) are hardly understandable, they seem to be created ad hoc (e.g., “epiotic eminence”, instead of “prominentia of the canalis semicircularis posterior”), thus difficult to use in anatomical comparisons.
- Literature seems to be well referenced & relevant. However, I must confess that I did not check this aspect in detail.
- Figures are relevant, high quality, but not well labelled & described (see below).

Experimental design

- Research question are well defined, relevant & meaningful. It is stated how the research fills an identified knowledge gap.

Validity of the findings

- The descriptive parts are written well, mostly with correct terminology.This is in contrast with figure plates, which sometimes include incorrect interpretations.
- The main purpose of this paper is to describe a new taxon, differentiate it from other related taxa, and present all its distinctive characters in a way that other researchers could easily use them in their comparisons. The crucial point in this process is Diagnosis. Because it is based on a single, incomplete (= fragmentary) specimen, the authors were able to use only a few (6) characters:
1. Skull hyperossified
2. Frontoparietals coossified, lacking midline suture
3. Frontoparietals covered by pit-and-ridge ornamentation
4. Frontoparietals bearing a smooth occipital flange
5. Frontoparietals without frontoparietal incrassation
6. Deep, groove-like central recess on the posterodorsal surface of braincase on each side of foramen magnum, housing the foramen of the occipital artery.
The first two are correlated with one another. No.3 is not clear – judging by Fig. 1A, it consists of robust tubercles of various sizes, but not of pits separated by ridges. Micro-CT reconstructions may not be accurate. Thus, can Fig. 1A be completed by a photograph of the ornamented surface? The frontoparietal incrassation in living anurans fill the fenestrae separated by the tecta in the roof of the braincase, which results in smooth ceiling of the braincase. In living anurans the incrassation can be recognized only after outlines of bone (of frontoparietal) and cartilage (of tecta). In hyperossified taxon, the borders between ossified structures disappear (as is the case with proc. paraoccipitales and prominent posterior semicircular canal), thus the braincase ceiling becomes smooth. The deeper central recess of the three recesses (the medial and lateral are shallower) is a character which may be subject of individual or developmental variation. Thus, only characters 4 and 6 may be considered reliable, which is not too much. In this situation, I would advice to strenghten the Diagnosis by inclusion of Differential diagnosis, which might better illustrate differences from other related taxa.

Additional comments

- Some anatomical interpretations are incorrect (e.g. “carotid canal”, which is actually the canal for the arteria occipitalis; curiously, it is given under correct name in the Diagnosis, line 218); anteriorly, where it leaves the frontoparietal it is called the arteria orbitonasalis, not “carotid artera” as is the case of explanations to Fig. 1; “foramen perilymphaticum” in Fig. 1G, which is actually the foramen endolymphaticum).
- Why all specimens were microCT scanned? Were they covered by sediment? This should be mentioned in Material and Methods section.

Questions and remarks to the authors
Line 33: I strongly recommand that the new taxonomic name is not introduced in the Abstract, which sometimes is published separately from the main body of the text; this may potentially create confusion for those who assemble records for synonymy. It is recommanded to use the new name first in the Systematic Paleontology section, i.e. in lines 190 and 193.
Line 65: Should not this be 100–95 Ma?
Lines 76-77: The sentence does not seem to be complete or is formulated incorrectly.
Line 81: The definite article should be removed.
Line 82: Should not be “of” inserted before “these fossils?
Lines 213-219, Diagnosis: Hyperossification is manifested not only by coalescence of the braincase with the frontoparietal, but also with parasphenoid. In the frontoparietal, the paraoccipital processes fuse without any distinctive suture on either side with the prominentiae of the posterior semicircular canal. Why do you call this structure of dual origin the eminentia epiotica?
More important - how did you recognize that the frontoparietal has no incrassation? In fact, incrassation(s) are dermal bone (frontoparietal) fillings in the fenestrae of the roof of the braincase; the fenestrae are separated by narrow bands of the endochondral bone. Thus, both dermal and endochondral bones together, make the roof of the braincase continuous, such that the frontoparietal incrassation cannot be recognized. Thus I do not consider this diagnostic character (absence of FRP incrassation) reliable.
You mention among diagnostic characters also the pit-and-ridge ornamentation. However, in Fig. 1A it rather consists of large but low tubercles on flat, horizontal surface. How reliable is your microCT reconstruction in this respect?
The three recesses on the postero-dorsal surface of the skull are insertion areas for muscles coming from the trunk (intertransversarii superiores; see Gaupp 1893, fig. 63). This area is usually highly variable, as may be judged also from comparisons of the left and right sides in Fig. 1E. Do you think that such weak diagnosis is sufficient for distinguishing the holotype from other similar material?
Occipital flange (ocfl) is mentioned among diagnostic characters, but it is not given among abbreviations of Fig. 1.
Line 230: Sing. processus paraoccipitalis, plural processūs paraoccipitales, but here it is better to use the anglicized form “paraoccipital processes”.
Line 231: precisely “prominentia ducti semicircularis posterioris”.
Lines 233 and 235: what is the difference between “lateral expansion of the frontoparietal” and “tectum supraorbitale”?
Lines 236-237: See above, Diagnosis.
Line 239: See above.
Line 244: It is the same artery, but from the point where it enters the orbit, it is called a. orbitonasalis (Gaupp 1899, figs. 82, 89).
Line 256: Crista parotica always extends laterally.
Line 259: Fig. 1B, G (?)
Line 269: In fact, partition between the braincase and otic capsule.
Lines 272-273: This variation is very common in anurans.
Line 276: It is important to write that your reconstructions in Fig. 2 represent casts of the cavities in the left otic capsule (elsewhere you use the term “endocast”, which is also understandable).
Lines 286-287 and Fig. 2: Given that you describe the endocast of the otic capsule, your perilymphatic cisterne contains also the sacculus and lagena, surrounded by closely adjacent wall of the perilymphatic space, but its more spacious part (i.e., the cisterne) is actually located only in the posteromedial portion of the otic capsule (according to Wever 2014; see also your line 292). So, that part of the endocast which is labeled “pycist” in your Fig. 2 is the cistern in posterior view (Fig. 2B) but it cannot be observed in anterior view (Fig. 2A), where is the sacculus located, which closely adjoins its wall. Furthermore, the “lateral chamber” is that part of the perilymphatic space, where its wall is in contact with the inner surface of the footplate and operculum, so it is a pitty that the contact surface between the perilymphatic space and these two middle ear structures are not involved in your reconstruction. In any case, be precise and describe this part as “transverse section through the perilymphatic space close to fenestra ovalis”, rather than “oval window” which is confusing. Finally, the same situation is in the opposite side, which is labeled “acoustic duct” (accdt). This is actually the transverse section through that part of the perilymphatic space bulging into the braincase cavity, which also contains the stato-acoustic nerve. Note also that the “round window” occurs only in mammals at the basal turn of the cochlea. I have not heard about it in amphibians where the auditory receptors are located beyond lagena.
Line 294: There are two little foramina in the lateral wall of the jugular canal called the foramen perilymphaticum superius and inferius (Stadtmüller 1936: Handbuch der vergl. Anatomie der Wirbeltiere, pp. 527-528, fig. 394), but never identified as “round window”.
Line 298: The correct name is the stato-acoustic nerve, because it contains fibres from both organs.
Line 302: This is correct, but see Fig. 1G, pf.!
Line 306-307: Please, can you include a brief information which terminology you follow? I do not understand what is “otic plate” (= “ramus paroticus”) and “posterior process” (of what?). Sanchiz (1998) tried to make a simple explanatory list of anatomical terms, but he forgot to include the squamosum! The aim of unified anatomical terminology is mutual understanding among anatomists, but this is not possible in current situation when various terminologies are used. I used the terminology of squamosum which is explained in Roček (1981, fig. 33).
Line 324: Posterior process of what?
Lines 326-328 and Fig. 3: Posterior margin (of what?) bears a shallow depression ... which is articular facet for the quadratojugal. However, the quadratojugal articulates only with the quadrate (sometimes also both coalesce together), and the quadrate has a suture with the ramus “posterior” of the squamosal, not with the processus posterior of the ornamented part of the squamosal (i.e., lamella alaris). This is an example of terminological mess.
What is sdp in Fig. 3 (not found in abbreviations), squvre (not found among labels in Fig. 3)?
Line 333: Thus, this “large” process is the proc. zygomaticomaxillaris (Figs. 3F, G)? Please, check the text in lines 334-341 with Fig. 3.
Line 365: What is “anterior vertebral canal”? Please, rewrite this sentence.
Line 376: Should it be “as wide as long”?
Line 380: “posteriorly” can be removed.
Line 387: One of ... vertebrae.
Line 428: What about Cretaceous of Utah (Roček et al. 2010)? Scotiophryne pustulosa Estes, 1969 ? Theatonius lancensis Fox, 1976 ? “Eopelobates sp.” from the late Maastrichtian of Wyoming and Montana (see Gardner 2008)?
Lines 456-457: This sentence is incomplete, please check.
Line 470: Arteria orbitonasalis.
Line 483: median suture
Lines 496, 199: posterior process of what on the posterior surface of the frontoparietal?
Line 501: See above.
Line 545: Better anglicized “paraoccipital processes”.
Line 556: its
Line 560: head or ball in your terminology?
Lines 568-569: shallow and poorly delimited fossa cubitalis is also in permanently water-dwelling Palaeobatrachus, probably due to stretched forelimbs in swimming, like in Xenopus.
Line 572: ulnar epicondyle
Line 586: iliac, because of Latin crista iliaca, not crista “iliala”. See also your line 597: inter-iliac.
Line 588: Do both these names mean the tuber superius?
Lines 592-593: Please rework this sentence, because its souds a little messy.
Line 596: ilioischiadic
Line 600: You can also include (or at least check Roček Z., Gardner JD, Eaton JG, Přikryl T. 2013: Anuran ilia from the Upper Cretaceous of Utah – Diversity and stratigraphic patterns. In: Titus AL and Loewen MA (eds) At the top of the Grand Staircase, pp. 273-294)
Lines 658-659: Why you use sometimes “otic plate” and sometimes “lamella alaris”? Terminology should be consistent.
Line 660: antorbitale is correct.
Lines 683-684: Palatine is present also in Early Cretaceous Genibatrachus (unpublished).
Line 749: Did anybody complete investigation of the material (mentioned as Pyxicephalus or Aubria) in Blackburn D., Roberts E. (2013) A late Oligocene anuran fauna from the Nsungwe Formation, southwestern Tanzania. Program and Abstracts of the 73rd Meeting of the Society of Vertebrate Paleontology. Supplement to the online JVP October 2013, p. 88?
Line 810: In Beceten is correct.
Fig. 1 A is correct; B is right lateral; C is ventral; should be B; E is posterior, slightly dorsal; left lateral is absent! F is anterior, not “left medial“; D is correct; G is correct.
Lines should be more precise (e.g., in F should reach up to the groove, jugular foramen in D should be indicated by arrow, because it is not visible, etc.). In explanations, Latin (e.g., tectum supraorbitale, pars contacta) and anglicized terms (prootic and majority of others) are mixed. cc = “carotid canal“ is actually canal for the arteria occipitalis; optic foramen is correct. pf = “perilymphatic foramen“ is actually endolymphatic foramen (always above acoustic foramina) in G. Two perilymphatic foramina are in the lateral wall of the foramen jugulare.
Fig. 2 Given that the pictures illustrate casts of the inner cavities of the otic capsule, the figure legend should be changed to: Inner casts of the left otic capsule. Please, use also precise anatomical terminology, which is in case of inner ear especially important. What you call the “oval window“ is actually transverse section of the sacculus; sacculus never expands in frogs. The canals are generally called the anterior, posterior, and lateral semicircular canals, which have an ampulla on one of their ends (thus, “ampulla of the anterior semicircular canal“, sups is crus commune, accdt is in fact canal for acoustic nerve, etc.). “Otic chamber“ and “round window“ can be recognized only in mammals, but not in frogs.
Brace below A in figure plate is a typing error?
Fig. 3 What is sdp? (absent in the list of abbreviations, but used in A). sqvre and sqvr are used in B, but only squvre is given among abbreviations. sqpp in D, but not among abbreviations. What is what? mr points into the recess, not to the ridge in H

Minor additional errors are highlighted in yellow in the annotated PDF version.

Reviewer 2 ·

Excellent Review

This review has been rated excellent by staff (in the top 15% of reviews)
EDITOR COMMENT
The reviewer pointed out many aspects of the morphological description that needed further review.

Basic reporting

Excellent

Experimental design

NA

Validity of the findings

Valid

Additional comments

Review


The introduction and background material are well-written and thorough.

Line 184: The sentence "We based this constraint tree (available in the
Supplemental Materials) primarily on recent phylogenomic analyses, including Feng et al (2017), Streicher et al. (2018), Yuan et al. (2018), and Hime et al. (2021)." is vague and needs specificity. Exactly which of these studies was the basis for the constraint tree. As written, it appear the constraint tree was constructed with no particular analysis, just a "feeling." Some of these studies do not support the same clades. I don't have any preference for one over the other, but the authors should state exactly how the constraint tree was constructed.

line 197: "posterior braincase preserving incomplete frontoparietals, parasphenoid,
198 and prooticooccipitals." As worded this implies that the frontoparietals and parasphenoid are part of the braincase, which is not correct. Also, I don't ever recall seeing the word "prooticooccipitals". Others have used otoccipitals or "prootic and exooccipitals." Frogs don't have a basioccipital or supraoccipital.

Also, Fig 1 labels the "prootic" not prooticoccipital. Use consistent terminology throughout.

line 216: The authors should describe somewhere the significane of "no incrassatio frontoparietalis on the ventral surface of the frontoparietals;"

line 233: pars contacta? Please state where the terminoloy comes from, not just where the list of characters is (Lemierre?)

line 244. The "occipital artery" foramen is mentioned, but this is labeled in Fig 1B as the carotid artery.

Fig 2. Is this the left or right inner ear? Also, the order of labeling A, B, C, is odd. Why not label the top three as A, B, C?
Also there is a large black brace that obscures one of the labels.

line 292: "In the posteromedial region of the perilymphatic cistern, a short and large canal, representing the perilymphatic duct, opens posteriorly (Fig. 2B, D) into the braincase and the condyloid fossa (i.e., “round window”; Wever, 1985)."
In Fig 2. the round window is clearly labeled with no ambiguity, but the use of "round window" in the text implies that this opening may not be the round window.

line 297 "housed the pathway of the cranial nerve VIII (Gaupp, 1896), representing the
acoustic nerve (Duellman and Trueb, 1994)." This is confusing. Are the authors subdividing VIII into acoustic and vestibular branches. Many refer to VIII simply as "acoustic" even though it has hearing and equlibrium functions.

line 310: "though it is not uncommon for anuran cranial bones to display variation in ornamentation within an individual". Not clear what you mean. Do you mean there is variation in ornamentation among different elements? This is not a surprise. Or do you mean something else?

line 346: "Although the shape of the centrum varies...". How does it vary? In cross section? Please be more specific.

line "In other anurans, there is documented variation in the length of the centra of...". Certainly Trueb (not cited) has noted this in several earlier papers.

Fig 4: postpostzygapophysis, not postzygapophyse. Same for pre-

Fig 4I. "spinal foramen" is labeled. As far as I know, no frog has a spinal foramen within the vertebra; the spinal nerve always exits between vertebrae. Some salamanders have intra-vertebral spinal foramina--see papers by Jim Edwards.
I don't know what this foramen might be.

In general, the heavy black scale bars are not esthetic and distract from the otherwise excellent figures. Can these be made thinner?

line 410. These for superfamilies are all crown groups, so you are not considering stemm-fossils that may have co-ossified skulls. Please re-write.

line 416: Where is Gobiatidae in your tree?

line 434: "However, the principal synapomorphies used to diagnose Neobatrachia, such as the presence of palatines (also called neopalatines in neobatrachians;
Baez et al., 2009) cannot be assessed based on the preserved elements of Cretadhefdaa."

A couple of points here. Why is the presence of palatines is more of a "principal" synapomorphy than the bicondylar sacrum?

However, the authors have unknowning documented a derived character state that does place the fossil at a lower level in the tree: the bicondylar sacrum. As far as I know, this is found only in Neobatrachia. I think this is worth more discussion.

line 480: "their attribution to Calyptocephalellidae is certain." Just a comment: I don't agree that it is certain. The tree support is not much stronger than many nodes in the tree in this paper. I think the authors of Calyptocephalella satan were influenced heavily by geography.

General taxonomic note: Hyloides, sensu Frost is not monphyletic because it includes Sooglossidae, a clear error. Hyloidea, which you do use in some parts of the ms, is the appropriate name.

Ranoides sensu Frost is the same as Ranoidea, which was used for at least 40 years before Frost. In the interest of stability, and following Feng et al., I suggest using Ranoidea (especially since Blackburn is a co-author of Feng et al.

The legend for Fig 7 does not indicate what the numbers on the tree branches mean.

---

## Round 0.2 · Minor Revisions

The same two reviewers commented on this new version. While R2 is happy with the changes made, R1 still has some discordance regarding the anatomical description. Make sure to prepare a revised version that takes into account all those comments.

·

Basic reporting

Introductory part good (including English), anatomical part still with errors.

Experimental design

No comment. This is a classical descriptive paleontology.

Validity of the findings

No comment.

Additional comments

Remarks to the authors
Those parts of the text highlighted in red deserve your revision.
Line 183: Correct to: Calyptocephalellidae
Line 184: Neoaustrana (?)
Line 186: Natatanua (?)
Line 232: differentiated
Line 241: median ... between
Line 250: prooticooccipitals
Line 257: posterior process (of the frontoparietals ?)
Lines 261-262: As I objected last time, in fully coossified tecta with thickened parts of the frontoparietals the ventral surface of the braincase roof will be smooth (i.e., the incrassations will not be recognizable).
Line 263: prooticooccipitals (ossified parts of the otic capsules) are two, like one on each side (as you mention in line 279).
Line 283: for (?; compare line 291).
Lines 325-326: “the large foramina of the medial wall of the braincase is the fusion of the two acoustic foramina”. This does not make sense. First, these foramina are in the lateral wall of the braincase; second, the large foramina cannot be the fusion of acoustic foramina. Please, reword the sentence.
Line 328: Again, this is the lateral wall of the braincase.
Line 334: Insert: “... the base of the processus posterolateralis”. This needs to be corrected in labeling in Fig. 3A-E.
Line 342: The lamella alaris in complete only on its anterior end.
Line 343: Remove “possibly a fused”. It is always part of the squamosum, like processus posterolateralis. Furthermore, it is true that the ramus paroticus adjoins the crista parotica, but this is the common scheme in all anurans, not only in Beelzebufo.
Lines 348-349: This is not true! The squamosum is always connected to the crista parotica by means of its ramus paroticus, disregarding whether its lamella alaris is in contact with the frontoparietal table or not.
Line 353: Processus posterolateralis lacks ornamentation in all frogs.
Lines 354-355: Totally incorrect.
Line 357: Middle portion would imply that there should be a part of the orbital margin, which is not true.
Line 360: See comments to Fig. 3 below.
Lines 366-367: This is incorrect, see below.
Line 406: to be elongated (?).
Line 444: to coalesce and to fuse mean the same.
Line 452: Pelobatoidea or Pelobatidae (?)
Line 455: This is highly improbable, not supported by any evidence.
Line 509: seem to be similar (?)
Line 570: Better: paraoccipital processes.
Line 587: Same as previous.
Fig. 1 ef is not explained. Endolymphatic foramen is erroneously given as pf in Abbreviations.
Fig. 3A-E squamosal. sdp is not explained. This set of pictures is confusing and partly incorrectly identified. The lamina alaris, if ornamented, always bears ornamentation on outer surface and is smooth on inner surface. Its anterior end (“sdp”) cannot be the processus posterolateralis, thus cannot bear the quadratojugal facet. The broken base of the processus posterolateralis is identified incorrectly as “sqvl”.
Fig. 3F-H maxilla. If pzm means processus zygomatico-maxillaris (i.e., at the posterior end of the orbit) and par ? means palatine articulation (at the level of the postnasal wall, where is the orbital margin that should be between the both? Moreover, the palatine, which is a flat, horizontal bone adjoining the postnasal wall ventrally, should be connected by a suture to the horizontal lamina, not to the oblique ridge on the inner surface of the vertical portion of the bone. What you identify as pzm is actually broken end somewhere close to proc. frontalis. What you indicate mr is the maxillary fossa (or recess) for the processus antorbitalis of the palatoquadrate bar. This maxillar fragment actually represents a short section at the level of the anterior margin of the orbit.
Fig. 5: Please explain white arrows.
Fig. 7 I cannot find red circle mentioned in the figure legend.

Reviewer 2 ·

Basic reporting

All criticisms were addressed satisfactorily.

Experimental design

All criticisms were addressed satisfactorily.

Validity of the findings

All criticisms were addressed satisfactorily.

Additional comments

None

---

## Round 0.3 · accepted · Accept

Thank you for making these final amendments to the manuscript in reply to the last comments of Reviewer 1. I think the paper is now much improved and ready to be published. Congratulations and thank you for choosing PeerJ as a venue for your interesting research.